

# Observations of biogenic volatile organic compounds over a mixed temperate forest during the summer to autumn transition

Michael P. Vermeuel[1, #], Gordon A. Novak[1, ^], Delaney B. Kilgour[1], Megan S. Claflin[2], Brian M. Lerner[2], Amy M. Trowbridge[3], Jonathan Thom[4], Patricia A. Cleary[5], Ankur R. Desai[4], Timothy H. Bertram[1*]

[1]Department of Chemistry, University of Wisconsin, Madison, WI, USA;
[2]Aerodyne Research Inc, Billerica, MA, USA;
[3]Department of Forest and Wildlife Ecology, University of Wisconsin, Madison, WI, USA;
[4]Department of Atmospheric and Oceanic Sciences, University of Wisconsin, Madison, WI, USA;
[5]Department of Chemistry and Biochemistry, University of Wisconsin – Eau Claire, WI, USA;

\# Now at Department of Soil, Water, and Climate, University of Minnesota – Twin Cities, St. Paul, MN, USA
^ Now at Cooperative Institute for Research in Environmental Sciences, University of Colorado Boulder, Boulder, CO 80309, USA and National Oceanic and Atmospheric Administration (NOAA) Chemical Sciences Laboratory (CSL), Boulder, CO 80305, USA

*Correspondence to: T.H. Bertram, timothy.bertram@wisc.edu

**Abstract.** The exchange of trace gases between the biosphere and the atmosphere is an important process that controls both chemical and physical properties of the atmosphere with implications for air quality and climate change. The terrestrial biosphere is a major source of reactive biogenic volatile organic compounds (BVOC) that govern atmospheric concentrations of the hydroxy radical (OH) and ozone ($O_3$). The oxidation of BVOC leads to the production of low-volatility products that can undergo homogenous nucleation or condense onto existing particles leading to formation and growth of secondary organic aerosol (SOA). Over forests, the net surface-atmosphere exchange of BVOC depends on the unique physiochemical properties of individual compounds as well as the mean physical conditions of the forest canopy that control surface emissions (e.g., temperature, sunlight, leaf area) and loss processes (e.g., uptake through stomata, surface adhesion). Here, we present measurements of BVOC mixing ratios and vertical fluxes over a mixed temperate forest in Northern Wisconsin during broadleaf senescence occurring in the summer-autumn transition. We use these observations to better understand the effects of changes in canopy conditions on net BVOC exchange. The BVOC investigated here include the terpenoids isoprene ($C_5H_8$), monoterpene hydrocarbons (MT; $C_{10}H_{16}$), a monoterpene oxide ($C_{10}H_{16}O$) and sesquiterpenes (SQT; $C_{15}H_{24}$), as well as a subset of MT oxidation products and dimethyl sulfide (DMS). During this period, MTs were primarily composed of $\alpha$-pinene, $\beta$-pinene, and camphene, where $\alpha$-pinene and camphene were dominant during the first half of September and $\beta$-pinene thereafter. We observed enhanced net MT emissions following the onset of leaf senescence, suggesting that senescence and abscission may be significant controls governing late season MT emissions in this ecosystem. We describe the impact of this MT emissions enhancement and shift in speciation on the potential to form highly oxygenated organic molecules (HOM). The calculated production rates of HOM and $H_2SO_4$, constrained by terpene





and DMS concentrations, suggest that biogenic aerosol formation and growth in this region should be dominated by secondary organics rather than sulfate. Further, we show that models using parameterized MT emissions likely underestimate HOM production, and thus aerosol growth and formation, during early autumn in this region. Further measurements of forest-atmosphere BVOC exchange during seasonal transitions as well as measurements of DMS in temperate regions are needed to effectively predict the effects of canopy changes on reactive carbon cycling and the relative contributions to aerosol production.

**1 Background**

Terrestrial ecosystems provide the largest source of reactive carbon to the global atmosphere, with emissions estimated to exceed 1,000 Tg yr$^{-1}$ (Guenther et al., 2012; 1995), greater than those of methane (~550 Tg yr$^{-1}$) (Saunois et al., 2016) and all anthropogenic volatile organic compounds (VOCs) (~200 Tg yr$^{-1}$) (Huang et al., 2017). More than half emitted biogenic VOC (BVOC) are in the form of reactive terpenes (isoprene, $C_5H_8$; monoterpenes, MT, $C_{10}H_{16}$;

sesquiterpenes, SQT, $C_{15}H_{24}$), which control oxidant loadings as well as the production rate of secondary organic aerosol (SOA) in select regions (Curci et al., 2009; Johnson and Marston, 2008; Lee et al., 2006b). The oxidation of terpenes generates highly oxygenated organic molecules (HOM) that can nucleate to form new particles or contribute to the growth of existing particles (Bianchi et al., 2019; Ehn et al., 2014; Fuentes et al., 2016; Jimenez et al., 2009). A second class of BVOC that can also contribute to aerosol growth and formation are reduced sulfur compounds (e.g.

dimethyl sulfide, DMS, $C_2H_6S$) (Lamb et al., 1987; Staubes et al., 1989; Fall et al., 1988; Kanda et al., 1995; Berresheim and Vulcan, 1992; Brown et al., 2015). DMS can be oxidized to $SO_2$ and then terminate as $H_2SO_4$ (Barnes et al., 2006), which can contribute to aerosol production. Although DMS is emitted and detected in low quantities in forests, only small steady-state concentrations of $H_2SO_4$ (~1 pptv) are required to generate significant particle nucleation rates (Kirkby et al., 2011) and new particle formation (NPF) events.


These organic and inorganic aerosol nucleation and condensation routes impact climate both directly by interacting with incoming solar radiation and indirectly by providing condensation nuclei that can alter cloud properties, and thus, Earth's albedo. Accurate estimates of BVOC emissions in chemical transport models (CTMs) are required to evaluate the impact of BVOCs on atmospheric chemistry. Emissions of BVOCs from plants are commonly calculated in CTMs

using land-type dependent emission factors, estimations of temperature and photosynthetically active radiation (PAR), and satellite-derived foliar density (Guenther et al., 2006; 2012; 1995). However, different plant species in the same land-type can emit specific terpene molecules at varying rates (Benjamin et al., 1996; Geron et al., 2000) and stresses such as drought, enhanced UV irradiation, extreme heat, herbivory, oxidative stress, and enhanced air pollution, among others, can modify ecosystem-level BVOC emissions (Peñuelas and Staudt, 2010; Loreto and Schnitzler, 2010).

Further, our understanding of the net exchange of BVOCs with soils and the forest floor remains limited (Trowbridge et al., 2020) although recorded magnitudes of soil and litter terpene flux is negligible compared to canopy-scale, plant-dominated fluxes (Greenberg et al. 2012). While emissions of DMS from soils in tropical and subtropical regions have



been shown to contribute considerably to observed DMS (Brown et al., 2015; Yi et al., 2010), few observations exist in other latitudes (Goldan et al., 1987; Lamb et al., 1987).


The surface-atmosphere exchange of BVOC in forested ecosystems is commonly measured during the growing season when leaf temperatures and foliar density (and thus emissions) are highest (Spirig et al., 2005; Acton et al., 2016; Isebrands et al., 1999; Laffineur et al., 2011; Janson, 1993). There are few studies that monitor BVOC exchange during seasonal transitions, particularly in northern temperate regions and mixed forests (Fuentes & Wang, 1999; Karl et al.,

2003). Several seasonal studies have been conducted at the SMEAR II coniferous boreal forest site in Hyytiälä, Finland. These studies have shown that in some ecosystems the decomposition of needleleaf litter along with other emissions from the forest floor (e.g., soils) can contribute to sustained and enhanced MT emissions, along with a seasonal change in MT speciation (and thus reactivity) (Aaltonen et al., 2011; Hakola et al., 2000; 2003; Hellén et al., 2006; Mäki et al., 2019). Dal Maso et al. (2005) recorded peak aerosol formation and growth events occurring in May

and September at the SMEAR II site suggesting an enhancement in BVOC emissions during seasonal transitions. Autumn peaks in the emissions of acetone and acetaldehyde have been observed in a mixed hardwood forest in Michigan, which was attributed to both senescing and decaying biomass (Karl et al., 2003). Observations of VOC fluxes from a plantation site showed that during the onset of leaf senescence and shortly thereafter, four different species of the deciduous *Populus* genus (e.g., aspens, cottonwood) exhibited a burst of oxygenated VOC (OVOC) and

MT emissions while isoprene emissions ceased (Portillo-Estrada et al., 2020). Seasonal changes to vegetation can also influence the emissions response to temperature as well as the speciation of emitted compounds. For example, over the course of one year, Helmig et al. (2013) observed seasonal deviations in both the MT profile and temperature response factor of six coniferous species. Together, these studies suggest that environmental and phenological factors affecting northern temperate forests during the summer to autumn transition are likely modulating ecosystem-level

BVOC dynamics in ways that are not accurately represented in current global models. Whether peak emissions occurring at the tree dormancy transition period are due to decaying, abscised leaves, or the senescence process of the attached leaf is unclear.

Here, we evaluate how seasonality affects atmosphere BVOC exchange during the summer to autumn transition

through a novel dataset, collected by a proton transfer reaction mass spectrometer (PTRMS) coupled to an online gas chromatography (GC) instrument, of the mixing ratios, net ecosystem fluxes, and speciation of key BVOCs over a northern WI mixed temperate forest during September 2020. During this time, trees were exposed to a wide range of temperatures, accumulated precipitation, and sunlight, as well as a steep change in canopy condition and leaf developmental stage (mature leaves, leaf senescence, and leaf abscission), all of which have been shown to modulate

the quantity, direction, and speciation of BVOC exchange. In addition, the mixed canopy allowed for concurrent observations of BVOC emissions from both coniferous species and deciduous species, with the potential to identify species-specific responses. Among the data collected are vertical fluxes and mixing ratios of key reactive terpenes (isoprene, MT, and SQT), of $C_{10}H_{16}O$, a presumed monoterpene oxide (MTO), as well as mixing ratios of other MTO and DMS. Net fluxes are compared to common temperature- and PAR-dependent parameterizations of BVOC



emissions to assess the suitably of such parameterizations during this period. Additionally, we use this chemical dataset, along with field meteorological data, to constrain a photochemical box model to evaluate the impact of seasonal effects on BVOC concentrations and speciation and on the production of HOM ($P_{HOM}$) and sulfuric acid ($P_{H_2SO_4}$) with implications for aerosol production.

This work focuses on understudied routes of BVOC emissions in a temperate mixed forest canopy during the summer to autumn transition to better improve our predictive capabilities of net ecosystem fluxes, concentrations of reactive carbon, and chemical rates that estimate the production of low volatility oxidized products. Results from this study suggest that the physical changes in this forest can strongly modify the net exchange of important BVOCs and need to be considered to predict the contribution of reactive carbon to atmospheric composition and aerosol production.

**2. Methods**

**2.1 Overview of Measurements at the WLEF Very Tall Tower in Park Falls, WI**

**2.1.1 Site Description**

The PEcoRINO (**P**robing **Eco**system **R**esponses **I**nvolving **N**otable **O**rganics) study consisted of chemical and meteorological observations over the Chequamegon-Nicolet National Forest (CNNF) at the WLEF-TV very tall tower

US-PFa Ameriflux site in Park Falls, WI (45.945ºN, 90.273ºW) (Davis et al., 2003) from 6-30 September 2020. The landscape surrounding the tower is composed of grasslands, woody wetlands, and deciduous and evergreen forests as determined from the National Land Cover Database (NLCD) 2016 (Homer et al., 2012). This location has been used for many EC studies concerning the role of surface heterogeneity on heat and carbon exchange (Desai et al., 2008, 2010, 2015; Xu et al., 2017; Bakwin et al., 1998), as well as a multi-institutional, intensive field campaign focused on

the role of atmospheric boundary layer responses to scales of spatial heterogeneity in surface-atmosphere heat and water exchanges (Butterworth et al., 2021). Recently, this site has been used to investigate the exchange of $O_3$ and formic acid and the role of in-canopy chemistry on observed fluxes (Vermeuel et al., 2021).

**2.1.2 Meteorological and $O_3$ measurements**

For the PEcoRINO study, routine US-PFa site measurements of 10 Hz wind speed and temperature (Model K Style Probe, ATI, Inc.) at 30 m were used, along with relative humidity (HMP-155; Vaisala) and solar irradiance (LI-190; LI-COR, Inc.) (Desai, 1996). Continuous 1 Hz measurements of $O_3$ mixing ratios using a photometric analyzer (Model 49i; Thermo Fisher) were made at a sampling height of 30 m through an inlet composed of Type 1300 Synflex drawing 30 standard liters per minute (SLPM) of ambient air. The photometric analyzer was calibrated by generation of a

calibration curve every three days using an $O_3$ calibration source (Model 306 Calibration Source; 2B Technologies).



### 2.1.3 VOC measurements

A high-resolution proton-transfer reaction time-of-flight mass spectrometer (HR-PTR-ToFMS) (Vocus; Aerodyne Research Inc. and Tofwerk AG) (Krechmer et al., 2018) made continuous 10 Hz measurements of VOCs at 30 m through a separate 1/2″ OD, 3/8″ ID perfluoroalkoxy alkane (PFA) inlet drawing between 25-30 SLPM of ambient air in order to maintain turbulent flow in the sampling line. The Vocus subsampled from the main inlet with a 5 SLPM bypass through a PFA tee located immediately in front of the Vocus capillary inlet into the instrument drift tube. The sample flow into the Vocus instrument was 100 sccm with the remaining bypass flow exiting to the pump. The sample inlet was constantly heated to 40 °C and was wrapped in aluminum foil throughout to avoid any potential inlet photochemistry. Attached to the front of the inlet was a PFA funnel wrapped in aluminum foil to avoid significant moisture draw during precipitation events. Spectra with a mass range of $m/Q$ 10-504 and a resolution of ~5000 $m/\Delta m$ were collected, allowing for highly resolved determination of peaks in the mass spectrum. 1474 peaks were integrated using the Igor Pro-implemented (WaveMetrics, Inc.) Tofware software package (Aerodyne Research Inc. and Tofwerk AG). A three-point calibration curve using a non-methane VOC (NMVOC) standard (Apel-Riemer Environmental, Inc.) and ultra-zero (UZ) air (AI UZ300, Airgas) was collected every 4 hours to record dynamic in-field calibration factors for select compounds. Calibration standards and concentrations are presented in **Table S1**. Calibrations were not added to the entire inlet but were rather introduced by overflowing the subsampling line. Calibration factors have been shown in lab studies to be insensitive to water content for the Vocus (Krechmer et al., 2018) and for this specific instrument (Kilgour et al., 2021). In the field these values were on average 900, 1500, 800, and 5000 cps ppbv$^{-1}$ for isoprene, MT (α-pinene), SQT (β-caryophyllene), and acetone, respectively and with coefficients of variation of less than 10% across species throughout the study.

A GC system designed for online atmospheric analysis (ARI GC; Aerodyne Research Inc.) was used to separate isomers of reactive compounds (e.g., MT) by coupling to the Vocus to create a GC-ToF-MS system. A previous version of the instrument is described in detail in Claflin, et al. (2020) but will be briefly described here. In the GC-ToF-MS system, sample air passes through a multi-stage thermal desorption preconcentration (TDPC) system (Aerodyne Research, Inc.) to collect and focus analyte species from ambient air before separation on the chromatographic column. The GC sample flow rate is controlled via mass flow controller (MFC) and held at 100 sccm for 10 min, resulting in a 1 L ambient sample per GC cycle. Before collection onto the TDPC, sample gases passed through a sodium sulfite (Na$_2$SO$_3$) oxidant trap to remove reactive gases, such as ozone, to reduce sampling artifacts that can occur at high mixing ratios (Helmig, 1997). After passing through the oxidant trap, the sample is then collected onto a multi-bed sorbent tube (Tenax TA/Graphitized Carbon/Carboxen 1000, Markes International) which is then purged with zero gas for 2 min to reduce the level of trapped water. After the post-collection purge, the sample is then transferred to a multi-bed, narrow bore, cold trap (Tenax TA/Carbopack X/Carboxen 1003, Markes International) for focusing before injection onto the GC column. Both the sample collection and focusing are conducted at sub-ambient (20 °C) temperatures through the use of a Peltier thermoelectric cooler. After focusing, the flow is then injected onto a GC column which then undergoes a programmed temperature ramp from 35-225 °C. The column used in this study resolves non- to mid-polarity VOCs including hydrocarbons, oxygenates, and some nitrogen and sulfur containing



compounds (MXT-624, Restek). For this study, the ARI GC was used to resolve $C_5$-$C_{12}$ hydrocarbons, DMS, and some oxygen-containing VOCs. For the majority of this work, the total chromatograph times were 10 minutes, which allowed for the full resolution of all MT species at this site. A subset of 14 sets of chromatograms were collected with a 20-minute chromatograph length to also speciate SQT and larger isomers, although this was not used for routine analysis as it was too time-demanding. Chromatogram peak areas were fitted using the Igor-implemented TERN software v2.2.9 (Aerodyne Research Inc.) (Isaacman-VanWertz et al., 2017, 2022) and the resulting values, in units of cts s extraction$^{-1}$ where cts are signal counts, were multiplied by the ToF extraction rate (24.4 kHz) to calculate quantifiable cts.

In the field, operation was divided between 10-Hz ambient collection solely through the HR-ToF-MS and collection *via* the GC-ToF-MS system, herein referred to as real-time (RT) and GC-Vocus sampling, respectively. The GC-Vocus collection routine is described in the **SI**.

## 2.2 Post-field calibrations of the RT- and GC-Vocus systems

Following the PEcoRINO study, experiments were performed to determine: 1) the effect of the inlet on potential irreversible loss of VOC to the inlet wall, 2) calibration factors for both the RT-Vocus and GC-Vocus, and 3) GC retention times for authentic standards. Experiments were performed under two conditions: one with standards added to the entire heated inlet line at a flow of 28 SLPM and one with standards added to a clean PFA line ~1 m in length. This allowed for comparison of the field inlet with a clean, short inlet as well as determination of post-field calibration factors. Calibrations of the RT-Vocus and GC-Vocus systems were performed by staged dilutions of a VOC standard with a mixture of 80:20 ultra-high purity (UHP) $N_2$:$O_2$, herein referred to as synthetic ZA. GC sample collection times were maintained at 10 minutes. The experiments showed negligible (<5%) loss of MT, SQT, isoprene, and acetone to the field inlet. Lab and field calibration factors of each system through addition to the short, clean line were within experimental error (5%) (**Fig. S1**). Since calibration factors of DMS, methanol ($CH_3OH$), and various monoterpene oxides ($C_9H_{14}O$, $C_{10}H_{14}O$, $C_{10}H_{16}O$, $C_{10}H_{16}O_2$, and $C_{10}H_{16}O_3$) were not collected in the field, post-field calibrations of DMS, methanol, nopinone ($C_9H_{14}O$), thymol ($C_{10}H_{14}O$), camphor ($C_{10}H_{16}O$), and cis-pinonic acid ($C_{10}H_{16}O_3$) were performed to determine RT-Vocus calibration factors as 3900, 97, 1700, 1000, 5100, and 460 cps ppbv$^{-1}$, respectively. The calibration factor for $C_{10}H_{16}O_3$ was applied to $C_{10}H_{16}O_2$ for the purposes of estimating concentrations in this study. The calibration factor of $\beta$-farnesene, the primary observed onsite SQT, was determined to be 800 cps ppbv$^{-1}$ which is the same as the field-determined $\beta$-caryophyllene. Field and laboratory determinations of GC-Vocus calibration factors were also consistent, with isoprene calibration factors of $5.4 \times 10^4$ (field) and $5.7 \times 10^4$ (post-field) cts ppbv$^{-1}$ and α-pinene calibration factors of $8.7 \times 10^4$ (field) and $7.5 \times 10^4$ (post-field) cts ppbv$^{-1}$. Comparison of RT-Vocus and GC-Vocus calibration factors shows the expected enhancement in sensitivity of nearly a factor of 60, which can be attributed to the sample collection time. Post-field GC calibrations of DMS show a similar enhancement factor, where the GC-Vocus calibration factor is $2.38 \times 10^5$ cts ppbv$^{-1}$.





To determine the retention time of potential isomers that were not included in the field NMVOC standard, qualitative experiments comprising direct, standard additions to the GC were performed (**SI**). The retention times (RT) of the above-listed MTOs, $\beta$-farnesene, and DMS were determined following this method. The retention times of unverified isomers and peak positions of unknowns was estimated using Kovats retention indices (RI). To do this, a library of known RTs were paired with their Kovats RIs acquired from the NIST database (Rostad and Pereira, 1986) to generate

a curve of RTs and Kovats RIs. Non-calibrated compounds could then be estimated by pairing their observed RT with their RI retrieved from the fit of RT vs RI (**Fig. S2**).

### 2.3 EC Flux Method Data Processing and Quality Control

Direct observations of trace gas fluxes were made using the eddy covariance (EC) method. As per the EC method, the

vertical flux of a compound, *C*, can be calculated as the covariance of the signal of *C* with vertical wind, *w*, within a period of *n* measurements (Stull, 1988):

$$F_C = \overline{w'C'} = \frac{1}{n}\sum_i^n (w_i - \overline{w})(C_i - \overline{C}) \tag{1}$$

Fluxes were calculated by Reynold's averaging (Eq. (1)) of 30-minute blocks of 10 Hz *C* and *w*. Flux uncertainties were determined through calculation of the flux limits of detection (LoD) for each flux calculation period as described

in Langford et al. (2015). LoD was calculated at the 95% confidence level, $1.96\sigma$ (standard deviation) of $f_X(t)$ of the outer 20 points within a 400-point lag time window centered around the average campaign maxima. Spectral corrections were performed to account for the high-frequency attenuation due sensor separation, inlet damping, and instrument response (Horst, 1997). This method is described in the SI and gave a 2-4% flux correction, on average, which was below the flux and measurement uncertainty and was therefore not applied. Also included in the SI is the

analysis of cospectra, calculations of cross-covariance to determine lags in response time, and measures of flux quality control to reject periods of low shear-driven turbulence, non-stationarity, and unphysical lag times (Horst, 1997; Wilczak et al., 2001; Foken and Wichura, 1996).

### 2.4 Parameterizations for Surface Emissions of BVOC

Estimates of BVOC emissions were performed based on parameterizations of the MEGAN (Model of Emissions of Gases and Aerosols from Nature) model (Guenther et al., 2012). Briefly, emissions of isoprene ($E_{iso}$) are parameterized as:

$$E_{iso} = \varepsilon_{iso} \cdot \rho \cdot C_L \cdot C_T, \tag{2}$$

where ε is the emission factor (EF) which represents emission of BVOC into a canopy at standard conditions, $C_L$ and

$C_T$ are factors that account for deviations in photosynthetic photon flux density (PPFD) and leaf temperature from standard conditions, and ρ is a parameter that accounts for loss within the canopy. We approximate emissions of MT and SQT as:

$$E_{MT,SQT} = \varepsilon_{MT,SQT} \cdot \rho \cdot [(1 - LDF) \cdot \gamma_{T,LIF} + LDF \cdot \gamma_{T,LDF}] \cdot \gamma_P, \tag{3a}$$

$$\gamma_{T,LIF} = e^{\beta(T_s - 297)}, \tag{3b}$$



where LDF is the light dependent fraction of emissions, $\gamma_{T,LIF}$ is the light-independent temperature activity factor dependent only on T, $\gamma_{LDF}$ is the light-dependent activity factor that depends on T and PPFD, and $\gamma_P$ is light activity factor. In Eq. (3b) $\beta$ represents a temperature scaling factor and $T_s$ is the leaf temperature (here approximated as air temperature). Factors were calculated according to Guenther et al., (1995; 2012). While the results of these parameterizations can carry a considerable amount of error (upwards of 210%) they are used in this analysis to directly

compare relative magnitudes and diurnal profiles in observed emissions. In the analysis presented here, we compare parameterized emissions from Eq. (3) with the observed flux.

**2.5 Box Modeling**

To derive chemical reaction rates and subsequent product formation from observations, a box model was constructed

with the Framework for 0-D Atmospheric Modeling (F0AM) (Wolfe et al., 2016) using the Master Chemical Mechanism (MCM) v3.3.1 (Jenkin et al., 2015). The oxidation of camphene was added to MCM using rates from Gaona-Colmán et al. (2017) for OH and $O_3$ oxidation and from Martínez et al. (1998) for $NO_3$ oxidation. In this model, SQT chemistry was assumed to be dominated by $\beta$-farnesene due to a match in Kovat retention index and the likelihood of emission of this compound from primary conifer species in the region (**Sect. 3.2**). Rate constants for the

OH- and $O_3$-initiated oxidation of $\beta$-farnesene were taken from Kim et al. (2011). Since no published rate constants for the $NO_3$-initiated oxidation of $\beta$-farnesene exist, we estimate this rate to follow that of $\beta$-caryophyllene which already exists in MCM, although the value used is highly uncertain (factor of five) based on the range of published SQT + $NO_3$ reaction rate constants (Shu & Atkinson, 1995). The production of HOM was calculated using lab yields of extremely low volatility organic compounds (ELVOC) from $O_3$- and OH-initiated terpene oxidation (Jokinen et al.,

2015, 2016). Since the HOM yields ($Y_{HOM}$) from the oxidation of $\beta$-farnesene is unknown, it is estimated in our model to be the same yields as $\beta$-caryophyllene. **Table S2** lists the yields used for this model. Due to the absence of studies on HOM formation from camphene oxidation, it is assumed that $Y_{HOM}$ for camphene are the same as $\beta$-pinene due to the presence of an exocyclic double bond in both compounds. It was also assumed that the $Y_{HOM}$ from $\beta$-farnesene + OH was the same as $\alpha$-pinene + OH. Since there is no existing $Y_{HOM}$ from $NO_3$-initiated oxidation of any of these

species, we set this to a low value of 0.001 for all species. Based on the range of values in **Table S2** the uncertainty on unknown $Y_{HOM}$ can be up to an order of magnitude. Although we make assumptions for $NO_3$ oxidation and subsequent HOM formation it is expected that the range of uncertainty from these values has a small comparable effect on oxidation and products relative to $O_3$ and OH. However, if there was a large onsite source of $NO_x$ (and thus $NO_3$) or sustained county road emissions, then $NO_3$-initiated oxidation in this region may have an impact on nocturnal

BVOC oxidation and subsequent aerosol production. The chemistry of DMS and subsequent $P_{H2SO4}$ is evaluated using the mechanism employed in Vermeuel et al. (2020b).

Meteorology ($T$, RH, pressure) and mixing ratios of DMS, $O_3$, methanol, and acetone were constrained by measurements. Other unmentioned chemical initial conditions follow those in Vermeuel et al. (2020b). Observed or

parameterized fluxes were used to constrain MT, isoprene, and SQT concentrations depending on the model run. A



model diel profile in OH based on published measurements in a northern temperate forest was used and scaled by observed solar radiation ($[OH]_{peak}=4.0 \times 10^6$ molecules cm$^{-3}$) to account for lower net $P_{OH}$ on cloudier days (Faloona et al., 2001). The model planetary boundary layer (PBL) height was based on a September 2019 diurnal profile from 40 km south of WLEF that peaks at 1.2 km during the day (Duncan et al., 2022). Emissions were divided by the PBL

height to provide source rates. Model $NO_x$ mixing ratios followed typical diurnal cycles and had an average of 200 pptv, which is an estimate but representative of prior autumnal temperate mixed forest measurements (Seok et al., 2013). The model was used to simulate chemistry for 06-30 September 2020 which included a model spin up of one day.

## 3. Results

### 3.1 Observations at Park Falls, WI in September 2020

### 3.1.1 Meteorology

The CNNF canopy experienced a variety of meteorological and physical (e.g., leaf stage, leaf area) conditions during the sampling period (**Fig. S4**). For example, there was a wide range of observed daytime maxima (7.9-26 ºC) in ambient temperature (**Fig. 1a**). Wind speed (**Fig. 1b**) generally peaked in the late afternoon (13-16 CDT) with an

average daytime value of 2.3 m s$^{-1}$ (average daytime maximum of 2.8 m s$^{-1}$) and a range in daytime maxima of 1.9-6 m s$^{-1}$. Winds primarily originated from the west (**Fig. 1c**) with large, abrupt changes in WD concurrent with low wind speeds, indicative of periods of a stable boundary layer. There were many precipitation events throughout the study (**Fig. 1d**), with 24 September onward experiencing many rainy, misty, and cloudy days. Also shown in **Fig. 1d** is the LAI product from the Moderate Resolution Imaging Spectroradiometer (MODIS) sensor onboard the NASA Terra

satellite for the pixel over the WLEF-TV site (Savtchenko et al., 2004). Throughout September, we also observed a decrease in LAI from 4 to 0.8 m$^2$ m$^{-2}$, indicating loss of leaves or declining leaf greenness throughout the month, both of which may be crucial in controlling the exchange of BVOC either via surface area required for emissions and/or deposition and uptake. Measurements of PPFD (**Fig. 1e**) indicate an attenuation of solar radiation in the last week of the study, suggesting increased cloud cover at the site during that time.





**Figure 1: Meteorology at the measurement site: a. temperature, b. wind speed, c. wind direction (WD), d. precipitation (black line) and LAI (grey dots), and e. photosynthetic photon flux density (PPFD).**

### 3.1.2 Mixing Ratios of Ambient Chemical Species

We first examine the impact of physical and meteorological changes that occurred during this period on the mixing ratios of reactive terpenes (MT, SQT, and isoprene), DMS, and $O_3$ (**Fig. 2**). **Fig. 2a** shows the time series of summed MT ($\Sigma$MT) (black line) as detected through the $MH^+$ ion $C_{10}H_{17}^+$ (m/Q 137.1325). Concentrations of $\Sigma$MT peaked in the evening due to late afternoon emissions and build-up thereafter due to reduced vertical mixing and oxidative



removal. We observe an increase in ΣMT concentrations following 21 September, indicative of either an increase in tree emissions at the later stage of the leaf cycle, a stronger contribution of forest floor emissions, a decrease in a chemical or physical sink at this stage in the study, or some combination of the three. From on-site visual assessment, senescence defined by changes in deciduous leaf color (and thus the end of the growing season) in this region generally began around 16 September and leaf abscission began around 21 September (**Fig. S4**). In addition, MODIS LAI decreased by more than half (from 4 to 1.5 $m^2$ $m^{-2}$) by 21 September, indicating a large portion of the region's leaf area losing greenness. Since this region is a mixed forest with species of varying lengths of developmental cycles, these assessments are approximations based on a few studied trees and may not be reflective of individual species that undergo mid- or late-autumn senescence. Following the beginning of leaf abscission of deciduous trees, high concentrations of MT were observed. Prior to 21 September, peak daily mixing ratios were regularly below 0.5 ppbv but were above 1.0 ppbv following 21 September, peaking at 1.4 ppbv. Average, concentration diel profiles for all species in **Fig. 2** for periods before and after 21 September are presented in **Fig. S5** to highlight these changes. The campaign-average [ΣMT] was 0.26 ppbv which is close to autumn measurements of [ΣMT] at the SMEAR II station in a boreal coniferous forest in Hyytiala, Finland (0.25 ppbv) where the latter would be expected to have a higher density of MT-emitting species (Hakola et al., 2003).

Data show that the speciation of MT changed throughout September (**Fig. 2a**). The colored regions in **Fig. 2a** show fractions of the major, identified MT isomers using the GC-Vocus. Prior to senescence, $\alpha$-pinene initially comprised on average ~40% of the total emissions, ~32% of total MT emissions during senescence, and decreased to ~20% during abscission. β-pinene showed a slight increase in MT fraction throughout the month, increasing from 44% to ~57% as leaves moved from the mature to abscission stages, respectively. Similarly, camphene also showed an increase in relative proportion shifting from mature (16%) to senescent (23%) stages. Speciation from the GC-Vocus is described more in **Sect. 3.2**.

**Figure 2b** shows mixing ratios of isoprene, as determined by the $C_5H_9^+$ ion (m/Q 69.06988), throughout the month of September. The $C_5H_9^+$ signal required correction due to the presence of *n*-aldehyde fragments in the $C_5H_9^+$ chromatogram. Although some of the *n*-aldehyde contribution in the chromatogram was determined to be from reactions of ozone with the system sorbent tubes due to unconditioned $Na_2SO_3$ used in the oxidant trap (**Section 3.2**), *n*-aldehydes were also observed in RT-Vocus measurements, and the known fragmentation of these n-aldehydes to $C_5H_9^+$ required a correction. Corrections were performed by taking advantage of the consistency in signal ratios of fragment ions to parent ions across the GC- and RT-Vocus (as in α-pinene, **Fig. S6**). To correct the $C_5H_9^+$ for *n*-aldehydes, the ratio of GC $C_5H_9^+$:parent ion peak area of the aldehydes of heptanal, octanal, and nonanal were multiplied by the parent ion signal to get the corresponding aldehyde $C_5H_9^+$ signal (**Fig. S7**). The sum of these *n*-aldehyde $C_5H_9^+$ signals were then subtracted from the total $C_5H_9^+$ signal to get an "isoprene-only" signal which were then calibrated for isoprene from in-field calibrations. The contribution of *n*-aldehydes was significant, making up 36% and 59% of the daytime and nighttime $C_5H_9^+$ signal, respectively. The daily peak in isoprene concentrations was variable, ranging from 0.13-1.1 ppbv, and the campaign average was 0.16 ppbv, a value between year-averaged



measurements of isoprene in a northern temperate forest in in MI in 2001 and 2002 (0.1 and 0.5 ppbv, respectively) (Karl et al., 2003). The two forests may not serve as direct comparisons, but comparison to the MI forest range does show the high interannual variability of isoprene in mixed northern temperate forests. **Figure 2c** shows the mixing ratios of ΣSQT detected at $C_{15}H_{25}^+$ (m/Q 205.1951) and calibrated for $\beta$-caryophellene. There is no clear diurnal cycle in ΣSQT and the campaign average [ΣSQT] was 7.2 pptv, a value over a factor of 6 lower than late summer

observations in a primarily coniferous forest (~44 pptv) where mixing ratios are expected to be higher and may serve as an upper bound (Bouvier-Brown et al., 2009).

Observations of DMS at CNNF, detected as $C_2H_7S^+$ (m/Q 63.0263) in the Vocus, are presented in **Fig. 2d**. The diel profile in DMS is consistent, displaying an evening maximum around 21:00 CDT and a minimum in the early morning

(~5:00-7:00 CDT). This profile of evening buildup is indicative of a compound that has a late afternoon source that extends into the evening. The short lifetime of DMS in the early morning suggests removal due to boundary layer mixing or advection since the lifetime of DMS against OH is too long to account for this loss (~1 day). DMS at CNNF is low, with an average mixing ratio of 7.7 pptv for the entire observation period. There was no dependence of [DMS] on leaf stage or LAI, suggesting that DMS may not be sourced from plants. Vertical mixing ratio profiles of terrestrial

DMS have been recorded at an Amazon Forest between September 2010 – January 2011, with mixing ratios <160 pptv. In that study, there was a clear enhancement of [DMS] in the late afternoon and at warmer temperatures and there was a strong nocturnal accumulation within the canopy and closer to the forest floor, indicative of light-independent soil emissions (Brown et al., 2015). Vertical distributions of DMS in a loblolly pine forest in Atlanta, Georgia also showed enhanced [DMS] closer to the forest floor and at night (~12 pptv) compared to the day (~4 pptv)

(Berresheim and Vulcan, 1992). The authors of the Atlanta study attribute this distinction to reduced photooxidation at night, and suggest that DMS emissions are from low-level wheat, which has been shown in lab to be a distinct source from soils (Fall et al., 1988). We believe observed DMS at Park Falls is due to soil emissions which, although highly dependent on microorganisms in the soil, have been proven to provide a modest source in other ecosystems (Goldan et al., 1987; Banwart and Bremner, 1975; Yang, 1996) and can explain observed mixing ratios at the site.


**Fig. 2e** shows half-hourly averaged mixing ratios for $O_3$ as measured by the photometric analyzer. The average [$O_3$] for the whole study was 23 ppbv and the day with highest measured [$O_3$] was 23 September 2020, a day that also experienced the peak in ambient temperature (26 °C). There was a period of sustained [$O_3$] maintaining >20 ppbv from 19-25 September, which correlates with periods of higher temperature in that time range.





**Figure 2: Mixing ratios of a. ΣMT, b. isoprene, c. ΣSQT, d. DMS, and e. O₃, from 06-30 September 2020. Leaf senescence generally began around 16 September and leaf abscission began around 21 September.**

Monoterpene oxides exhibited enhancements in ambient mixing ratio during the seasonal transition. Mixing ratios of the monoterpene oxide $C_{10}H_{16}O$, detected as $C_{10}H_{17}O^+$ (m/Q 153.1638) and calibrated for as camphor, exhibited a similar behavior to the time series of [ΣMT] (**Fig. 3a**). Following 21 September, mixing ratios are enhanced on average by over a factor of 2 compared to the period before 21 September. This suggests that $C_{10}H_{16}O$ and ΣMT follow the same mechanisms of primary emissions and/or that $C_{10}H_{16}O$ is a MT oxidation product. This type of observation from





a mixed temperate forest has been published before: observations from a mixed forest in New England show that summertime emissions of $C_{10}H_{16}O$ were closely related to those of ΣMT, although emissions were negligible by

September (McKinney et al., 2011). A recent study using a Vocus PTR-ToF-MS system at the coniferous Landes Forest found the diel profile of MT to be consistent with $C_{10}H_{17}O^+$, with an average evening $[C_{10}H_{16}O]:[ΣMT]$ of 0.03, compared to 0.08 in this study. The authors suggested that $C_{10}H_{16}O$ was partially sourced from directly-emitted camphor or a MT oxidation product (Li et al., 2020, 2021), although at the CNNF site there are few species that directly emit camphor and those that do (north white cedar, white spruce), emit camphor in low amounts (Helmig et

al., 1999). **Figure 3** provides concentrations of MTOs measured with the RT-Vocus at a collection of other ions that were determined to be lightly oxidized products of MT oxidation ($C_9H_{15}O^+$, $C_{10}H_{15}O^+$, $C_{10}H_{17}O_2^+$, $C_{10}H_{17}O_3^+$). The $C_9H_{15}O^+$ ion (m/Q 139.1117) is commonly assigned as nopinone, one of the main products formed during $β$-pinene ozonolysis (Atkinson and Arey, 2003; Lee et al., 2006a). In this study the ion was calibrated for nopinone, and the identity confirmed by GC observations (**Sect. 3.2**). Nopinone had the highest recorded concentration among the

observed monoterpene oxides. Other common assignments of observed monoterpene oxides are the major $α$-pinene ozonolysis product pinonaldehyde ($C_{10}H_{16}O_2$) which can appear as a parent ion (m/Q 169.1223; $C_{10}H_{17}O_2^+$) or as a fragment (m/Q 151.1118; $C_{10}H_{15}O^+$), and pinonic acid (m/Q 185.1172; $C_{10}H_{17}O_3^+$), a minor product of $α$- and $β$-pinene ozonolysis (Atkinson and Arey, 2003). The $C_{10}H_{15}O^+$ ion was identified by GC measurements to be primarily thymol but $C_{10}H_{17}O_2^+$ and $C_{10}H_{17}O_3^+$ could not be identified due to retention times outside of our collection window.

All monoterpene oxides in **Fig. 3** show an enhancement in concentrations following 21 September suggesting either a similar physical mechanism in the enhancement of emissions or increases due to the oxidation of concurrently increased MT. Concentration diel profiles for these monoterpene oxides for periods before and after 21 September are presented in **Fig. S8** to highlight changes due to the seasonal transition.



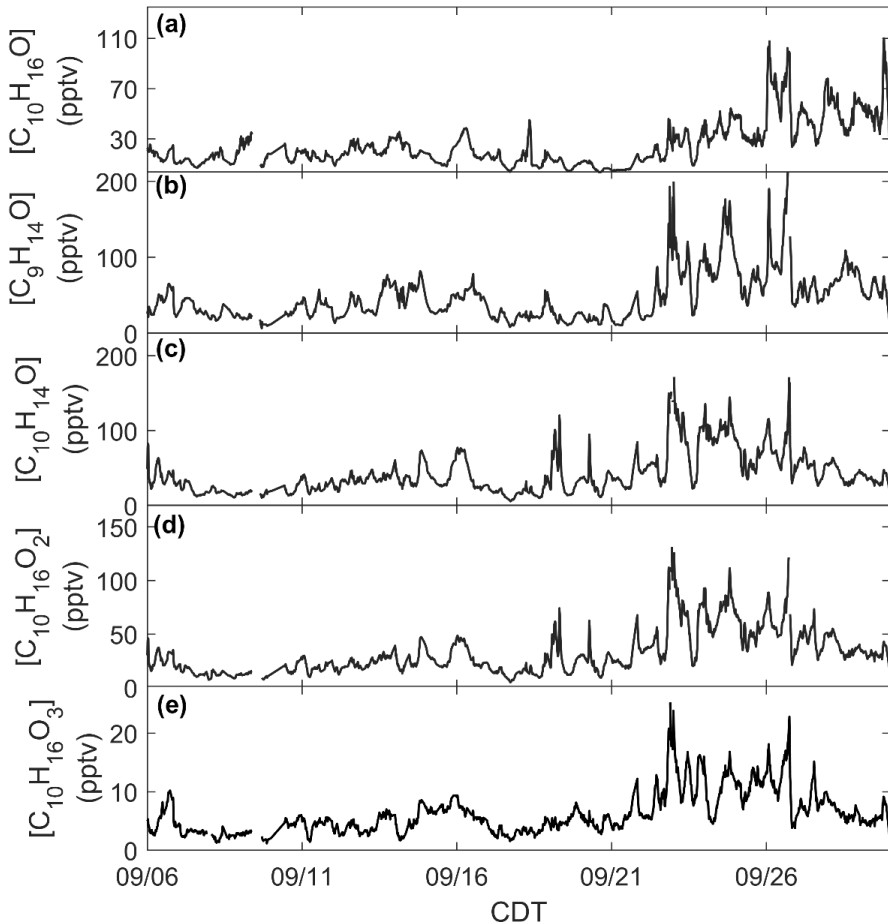

**Figure 3: Time series of monoterpene oxide concentrations: a. $C_{10}H_{16}O$ calibrated for camphor, b. $C_9H_{14}O$ calibrated for nopinone, c. $C_{10}H_{14}O$ calibrated for thymol, d. $C_{10}H_{16}O_2$ and e. $C_{10}H_{16}O_3$ calibrated for cis-piconic acid.**

### 3.1.3 Eddy Covariance Fluxes of BVOC

The effect of the seasonal transition on forest-atmosphere exchange of BVOC is shown in **Fig. 4** which presents the quality-controlled fluxes of $\Sigma$MT, $C_{10}H_{16}O$, $\Sigma$SQT, and isoprene. $F_{\Sigma MT}$ (**Fig. 4a**) regularly exhibited daytime maxima less than 1.0 ppbv cm s$^{-1}$ (195 $\mu$g m$^{-2}$ h$^{-1}$) prior to 21 September (Fig 4a). Following 21 September, emissions were enhanced (maximum 3.7 ppbv cm s$^{-1}$; 722 $\mu$g m$^{-2}$ h$^{-1}$) and in agreement with observed mixing ratios during the same time period. There was no strong increase in temperature during this period (Fig 1a), indicating that physical factors other than leaf temperature control emissions of $\Sigma$MT following leaf senescence. While there is a source area shift for $F_{\Sigma MT}$ from the west half to primarily southwest (**Fig. S9**) for pre- and post-21 September, respectively, it is unclear if this shift caused emissions enhancements since both footprints overlap. The diurnal profile of $F_{\Sigma MT}$ (**Fig. 4e**) shows that emissions follow a temperature profile, with emissions peaking in the late afternoon (13-15 CDT). The only other measurement of $F_{\Sigma MT}$ in this region was in July 1993 with values ranging from 37-740 $\mu$g m$^{-2}$ h$^{-1}$ based on leaf level



measurements and estimates of area foliage densities (Geron et al., 1994; Isebrands et al., 1999). Based on a synthesis of MT speciation in the US, a high end estimate of $F_{\Sigma MT}$ in Northern WI would be 935 $\mu$g m$^{-2}$ h$^{-1}$ (Geron et al., 2000).


The fluxes of $\Sigma$MT and $C_{10}H_{16}O$ are closely related, with $F_{C_{10}H_{16}O}$ enhanced by nearly a factor of 2.5, on average, following 21 September (**Fig. 4b**). A regression of $F_{C_{10}H_{16}O}$ and $F_{\Sigma MT}$ provides an r$^2$ of 0.83 and a slope of 0.041 ($F_{C_{10}H_{16}O}$: $F_{\Sigma MT}$) showing that the two processes are correlated (**Fig. S10**). Branch enclosure measurements of ponderosa pine trees at Manitou Forest in CO, USA from 21 August – 04 September 2008 showed consistent emissions

ratios of $C_{10}H_{16}O$:MT of ~0.1 (Kim et al., 2010) suggesting direct emissions of $C_{10}H_{16}O$. The New England canopy-scale flux study over a mixed temperate forest recorded a summertime $C_{10}H_{16}O$:MT of ~0.03 (McKinney et al., 2011), highlighting the range in this ratio depending on the observed plant types and ecosystem or potentially due to physical in-canopy loss of $C_{10}H_{16}O$. However, in the McKinney et al. study, $F_{C_{10}H_{16}O}$ and $F_{\Sigma MT}$ were not correlated (r$^2$=0.18).

$F_{isoprene}$ exhibited high day-to-day variability (**Fig 4c**; daytime maxima: 0.27-7.6 ppbv cm s$^{-1}$ or 27.4-772 $\mu$g m$^{-2}$ h$^{-1}$) and was not enhanced after 21 September, consistent with cessation of *in situ* leaf synthesis. The variability in $F_{isoprene}$ was partially controlled by ambient temperature, as observed by $F_{isoprene}$ enhancement on warmer days (e.g., 06 September; 22-23 September) and suppression on colder days (e.g., 08-09 September). The diurnal cycle of $F_{isoprene}$ (**Fig. 4g**) peaked with air temperature and was low or zero outside of daylight hours, implying that parameterizations

of isoprene emissions based on sunlight and temperature are appropriate during this season and that the covariance in $w$ and the signal of $C_5H_9^+$ is primarily from isoprene emissions and not of *n*-aldehydes. In addition, there was no measurable flux from the parent masses of heptanal, octanal, and nonanal, which further supports this conclusion. Previous area-averaged fluxes in this region from the July 1993 study, where leaf temperatures reached 35$^o$C and caused high emissions, were 1.13 mg m$^{-2}$ h$^{-1}$ (11.1 ppbv cm s$^{-1}$), providing and upper bound for the observations here.


**Figure 4d** presents, to our knowledge, the first canopy-scale fluxes of SQT in a mixed temperate forest. Similar to $F_{isoprene}$ and $F_{\Sigma MT}$, $F_{\Sigma SQT}$ also demonstrated a diel temperature dependence although the day-to-day variability was not as pronounced (**Fig. 4d**). In addition, the time series of $F_{\Sigma SQT}$ was not dependent on leaf stage, and we did not observe an enhancement in emissions post-21 September. Daily maxima of $F_{\Sigma SQT}$ ranged from 1.1-7.2 pptv cm s$^{-1}$ (0.366-2.19

$\mu$g m$^{-2}$ h$^{-1}$). No measurements of $F_{\Sigma SQT}$ have been performed near this site for comparison although branch enclosure measurements of summertime north temperate pine suggest canopy-scale SQT emissions up to as much as 300 $\mu$g m$^{-2}$ h$^{-1}$ (Holzke et al., 2006), providing an upper bound nearly two orders of magnitude larger than observations. Depending on the chemical lifetime of the dominant species in observed SQT, the magnitude and profile of $F_{\Sigma SQT}$ can be influenced by in-canopy ozonolysis. A study measuring above- and within-canopy ambient concentrations of SQT

in the Amazon showed that 46%-61% of SQT by mass undergo in-canopy ozonolysis (Jardine et al., 2011) and a multi-layer gas dry deposition model using observations from the SMEAR II station showed that ~70% of SQT is removed within the canopy due to chemical oxidation. We estimate the impact of within-canopy ozonolysis on $F_{\Sigma SQT}$ in **Sect. 4.1**.





The error bars presented in **Fig. 4a-d** were determined through calculations of flux LoD (**Sect. 2.3**). The campaign average uncertainties for MT, $C_{10}H_{16}O$, isoprene, and SQT were 25, 33, 27, and 37%, respectively. **Table 1** provides a summary of observed mixing ratios and fluxes through the study.

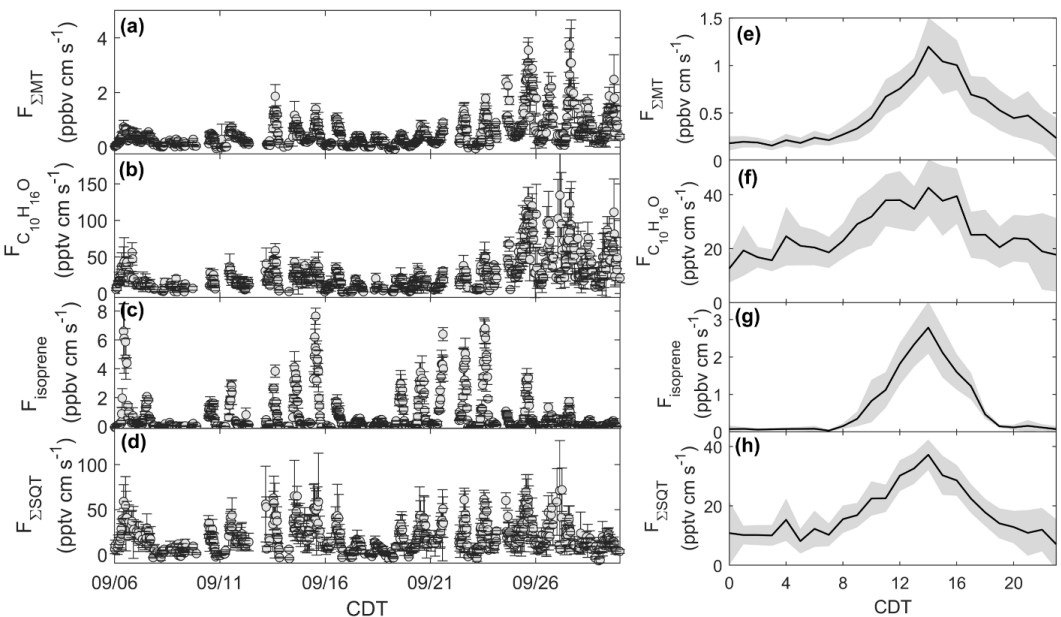

**Figure 4: Observed fluxes of a. ΣMT, b. $C_{10}H_{16}O$, c. isoprene, and d. ΣSQT from 06-30 September 2020. Also presented are diel profiles of e. ΣMT, f. $C_{10}H_{16}O$, g. isoprene, and h. ΣSQT. Shaded regions are 95% confidence intervals.**


| Molecule | MR Daily Peak Range (ppbv) | MR Mean (ppbv) (full study/ pre-09/21/ post-09/21) | *F* Daily Peak Range (ppbv cm s⁻¹) | *F* Mean (ppbv cm s⁻¹) (full study/ pre-09/21/ post-09/21) |
|---|---|---|---|---|
| **ΣMT** | 0.11–1.2 | 0.26<br>0.15<br>0.41 | 0.23–3.7 | 0.53<br>0.30<br>0.85 |
| **$C_{10}H_{16}O$** | $5.0 \times 10^{-3}$ – 0.11 | 0.026<br>0.017<br>0.039 | 0.010 - 0.14 | 0.028<br>0.017<br>0.042 |
| **Isoprene** | 0.10–1.5 | 0.16<br>0.14<br>0.20 | 0.27–7.6 | 0.84<br>0.86<br>0.82 |
| **ΣSQT** | $4.8 \times 10^{-3}$ – 0.027 | $7.2 \times 10^{-3}$<br>$6.5 \times 10^{-3}$<br>$8.2 \times 10^{-3}$ | $1.1 \times 10^{-3}$ – 0.072 | 0.019<br>0.018<br>0.021 |
| **DMS** | $7.6 \times 10^{-3}$ – $1.8 \times 10^{-2}$ | $7.7 \times 10^{-3}$<br>$7.7 \times 10^{-3}$<br>$7.8 \times 10^{-3}$ | -- | -- |
| **$O_3$** | 18–51 | 23 | -- | -- |



| | | | | |
|---|---|---|---|---|
| | | 22 | | |
| | | 26 | | |
| $C_9H_{14}O$ | 0.027-0.21 | 0.048 0.033 0.068 | -- | -- |
| $C_{10}H_{14}O$ | 0.020-0.17 | 0.043 0.030 0.061 | -- | -- |
| $C_{10}H_{16}O_2$ | 0.015-0.13 | 0.033 0.021 0.048 | -- | -- |
| $C_{10}H_{16}O_3$ | $3.0 \times 10^{-3}$ - 0.025 | $6.3 \times 10^{-3}$ $4.4 \times 10^{-3}$ $9.0 \times 10^{-3}$ | -- | -- |
| Methanol | 1.6-16.6 | 4.6 3.8 5.7 | 1.8-13.1 | 0.60 0.80 0.34 |
| Acetone | 0.72-3.3 | 1.3 1.1 1.5 | 0.10-0.99 | -0.023 0.012 -0.071 |

**Table 1: Summary of mixing ratios (MR) and fluxes (F) of select compounds during the PEcoRINO study.**

### 3.2 GC-Vocus Observations of BVOC

Use of the GC-Vocus allowed for speciation of MS peaks into the isomers that contribute to the total signal of product
ions. Throughout this section, GC field observations are either compared to retention times (RetT) of field calibrated
compounds (**Fig. S11**) or retention indices (RI) for compounds not directly calibrated in the field that required post-
field calibrations (**Fig. S12**).

**Figure 5a** shows an example chromatogram of $C_{10}H_{17}^+$, the MT product ion. There were three major peaks in the
$C_{10}H_{17}^+$ chromatogram (**Fig. 5a**): α-pinene (RetT = 488 s, $RI_{fit}$ = 922), camphene (RetT = 504 s, $RI_{obs.}$ = 942), and β-
pinene (RetT = 523 s, $RI_{obs.}$ =967) where $RI_{obs.}$ is the observed RI from the experimentally-determined fit of RetT vs.
RI. All other peaks in the chromatogram accounted for <5% of the total peak area and were thus considered negligible.
Due to the ubiquity of α-pinene and β-pinene from tree and forest floor emissions data, we hypothesized that these
compounds would be present at this forest. Camphene was also expected, as it has been observed from branch
enclosure measurements of seven tree species common to CNNF (e.g. yellow birch, aspen, fir, pine, and spruce
species) at a site approximately 80 km southwest of Park Falls, WI in Rhinelander, WI (Helmig et al., 1999). The α-
pinene peak was identified in-field using the NMVOC calibration cylinder (**Fig. S11**), the β-pinene peak was identified
post-study with a second NMVOC calibration cylinder, and camphene was identified post-study by flowing ZA over
solid camphene and collecting a chromatogram (**Fig. S12**). **Figure 6a** shows the time series of RT-Vocus ΣMT (black
line) and GC-Vocus ΣMT as the sum of α-pinene, camphene, and β-pinene peak areas. A regression of the two time
series (**Fig. 6c**) shows excellent agreement between the two methods (slope = 0.80, $r^2$ = 0.87), highlighting that the
three monoterpenes make up the majority of the $C_{10}H_{17}^+$ at this site in September.



**Figure 5b** shows the chromatogram of $C_5H_9^+$ from representative daytime (black solid line) and nighttime (red dashed line) collections. The RetT of isoprene was confirmed through in-field calibrations (**Fig. S11**). Both day and night chromatograms show isoprene contributing to a small amount (<10%) of the total signal, with major contributors being octanal (RetT = 549; $RI_{obs.}$= 999; $RI_{lit}$ =1001) and nonanal (RetT = 609; $RI_{obs.}$ = 1075; $RI_{lit}$ = 1102) (Adams et al., 2006; Merle et al., 2004). However, there is a clear distinction between daytime and nighttime isoprene while the other peaks remain relatively unchanged. This suggests the majority of the non-isoprene species contributing to $C_5H_9^+$ signal in this chromatogram are formed from reactions of the sorbent tube material with ambient $O_3$ since these products are well known to form this way (Lee et al., 2006). To confirm this, a post-field lab study was performed by sampling ZA containing 0-30 ppbv of $O_3$ through the GC-Vocus with and without the $Na_2SO_3$-filled oxidant trap. This experiment showed that a significant amount of $n$-aldehydes are produced from the sample trap at ambient $O_3$ (15-30 ppbv) in the absence of the oxidant trap (**Fig. S13**). An oxidant trap with unused $Na_2SO_3$ was then purged with $N_2$ and heated to 50° C for 1 hour before being placed in-line with the GC-system to remove any compounds adhering to the $Na_2SO_3$ powder surface. Results from this experiment showed that the background signal from octanal is high when sampling without the oxidant trap in ambient $O_3$ and increases with increasing $O_3$. When replacing the trap there is a hysteresis in signal from the traps, with signals decaying with each run after replacing the oxidant trap material. Because of this, the first GC series performed after oxidant trap replacement were discarded from analysis to clear out contamination from this. To check if $n$-aldehyde signal was from ozonolysis of compounds on the inlet surface, a post-field experiment was performed where varying concentrations of $O_3$ was added to the inlet manifold under field conditions with and without the oxidant trap (**Fig. S14**). This experiment showed that inlet ozonolysis has a small effect on detected $C_5H_9^+$, with about 1.5 cps of $C_5H_9^+$ generated per ppbv $O_3$ added. Since we were able to calculate some non-isoprene $C_5H_9^+$ signal (**Sect. 3.1**), we can conclude that there was a relatively small amount of non-isoprene species either produced in the gas-phase or from surface inlet reactions, but the majority of the GC non-isoprene $C_5H_9^+$ signal is from reactions of ozone with the sorbent tube. For future use of this GC-Vocus system, we suggest regularly replenishing the oxidant trap when enhanced $n$-aldehyde peaks are observed, using $Na_2SO_3$ conditioned just prior to use.

The time series of isoprene from the RT-Vocus (corrected for the n-aldehyde contribution to $C_5H_9^+$ and calibrated for isoprene) and the GC-Vocus (from the calibrated isoprene GC peak area) is shown in **Fig. 6b** along with a regression of RT-Vocus vs. GC-Vocus isoprene (**Fig. 6d**). There is good agreement between the data with an $r^2$ of 0.72, implying that our $C_5H_9^+$ correction method is a viable solution for calculating only isoprene from $C_5H_9^+$ and would be useful for other studies where fragments may contribute to a portion of a signal (e.g., correcting fragments out of $C_2H_5^+$ to derive butene). Since it is expected that the delivered calibrant varies by +/- 30%, the slope of RT- to GC-Vocus concentrations (0.70) is within the uncertainty of our calibration.

Only one peak (RetT = 900 s) was recorded in the chromatogram for $C_{15}H_{25}^+$ (**Fig. 5c**). The lack of additional peaks may be due to four factors: 1) other SQT isomers in too low of concentrations, 2) a low GC resolution at this retention time ($FWHM_{\beta-caryophyllene}$ = 9.5 s; $FWHM_{\alpha-pinene}$ = 2.4 s), where FWHM is the full width of the peak at half maximum,



3) isomers requiring elution times beyond the extended recording time of 20 minutes, or 4) condensation or irreversible loss of SQTs within the lines of the GC system. In-field calibrations show this peak overlapping with, but not exactly matching, $\beta$-caryophyllene (**Fig. S11**). The recorded RetT in the later collection times did observe a larger shift (+/-10 s) between collections and the lower resolution at this RetT which may lead to these discrepancies. The peak
recorded at this RetT may also belong to $\alpha$-cedrene, $\alpha$-humulene, or $\beta$-farnesene based on a matches in Kovatz RI ($RI_{obs.} = 1444$; $RI\alpha_{-cedrene,lit} = 1433$; $RI\alpha_{-humulene,lit} = 1454$; $RI\beta_{-farnesene,lit} = 1458$) (Yousefzadi et al., 2011; Medina et al., 2005). Based on the predominant conifer species at CNNF (red pine, white pine, and grey pine) (Haugen et al., 1998) it is not expected that $\beta$-caryophyllene, $\alpha$-cedrene, or $\alpha$-humulene would exist in large amounts. A review of SQT emissions from vegetation (Duhl et al., 2008) shows that for white pine, $\beta$-caryophyllene , $\alpha$-cedrene, and $\alpha$-humulene
make up 7%, 1%, and 5% of SQT emissions with the majority from $\alpha$-farnesene (57%). For red and grey pine, $\beta$-caryophyllene, $\alpha$-cedrene, and $\alpha$-humulene make up 0% of emissions with $\alpha$-farnesene ($\beta$-farnesene) making up 55% (41%) and 20% (77%) of red and grey pine emissions, respectively. Since no peak was observed for $\alpha$-farnesene ($RI\alpha_{-farnesene,lit} = 1509$ would produce a $RetT\alpha_{-farnesene}$ of 950 s), (Marongiu et al., 2003) we expect that the SQT detected onsite was $\beta$-farnesene. This peak RT was confirmed to be $\beta$-farnesene through lab additions of a mixture of farnesene
isomers to the GC. There was a small, shifting background in the RT = 890-910 s range (**Fig. S15**), although the magnitude of this peak was an order of magnitude lower than ambient collections. Since only 14 sets of 20-minute collections were recorded, all prior to 21 September, we cannot conclude if there was a change in speciation in $C_{15}H_{24}$ following leaf senescence.

A representative chromatogram for $C_2H_7S^+$ is presented in **Fig. 5d**. The major peak recorded in this chromatogram is DMS at an RetT of 90 s ($RI_{obs.} = 420$), which was confirmed through post-field calibrations (**Fig. S12**) and a second minor peak is from peak-fitting contamination of $C_2H_7O_2^+$ ($C_2H_7O_2^+$ m/Q 63.044; $C_2H_7S^+$ m/Q 63.02), tentatively ethylene glycol, at 308 s. This is confirmed through overlap at this RetT with a corresponding peak at 308 s in $C_2H_7O_2^+$ in background chromatograms (**Fig. S15**) and the signals of $C_2H_7S^+$ and $C_2H_7O_2^+$ are resolved for ambient Vocus
collections and result in two distinct time series (**Fig. S16**). Based on the collected chromatogram window, we assume that there only exists one isomer at $C_2H_7S^+$ which is DMS, although the RT-Vocus overestimates concentrations by 50%, on average (**Fig. S17**). This overestimation may be due to larger, late eluting species that fragment to $C_2H_7S^+$ but are not detected within our chromatogram window, or from species that cannot be resolved with the current column.



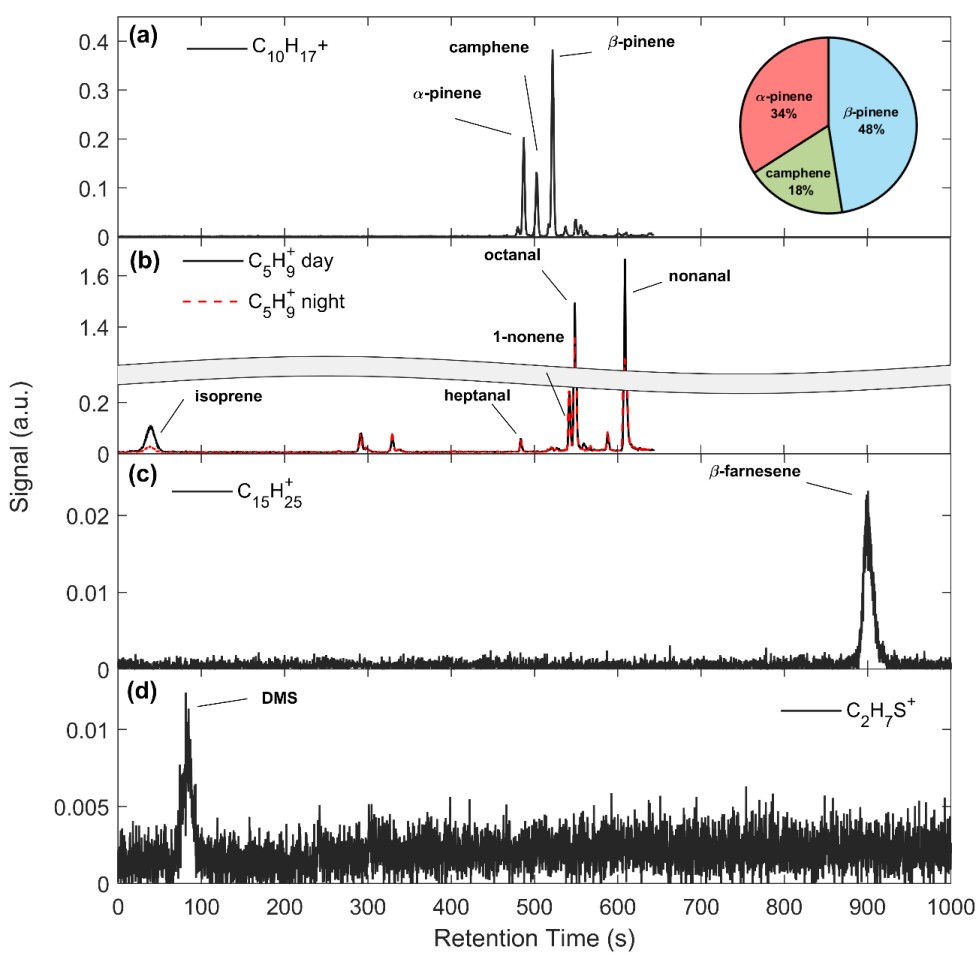

**Figure 5: a. Representative chromatograms of a. $C_{10}H_{17}^+$, b. $C_5H_9^+$, c. $C_{15}H_{25}^+$, and d. $C_2H_7S^+$ with identified isomers labeled.**



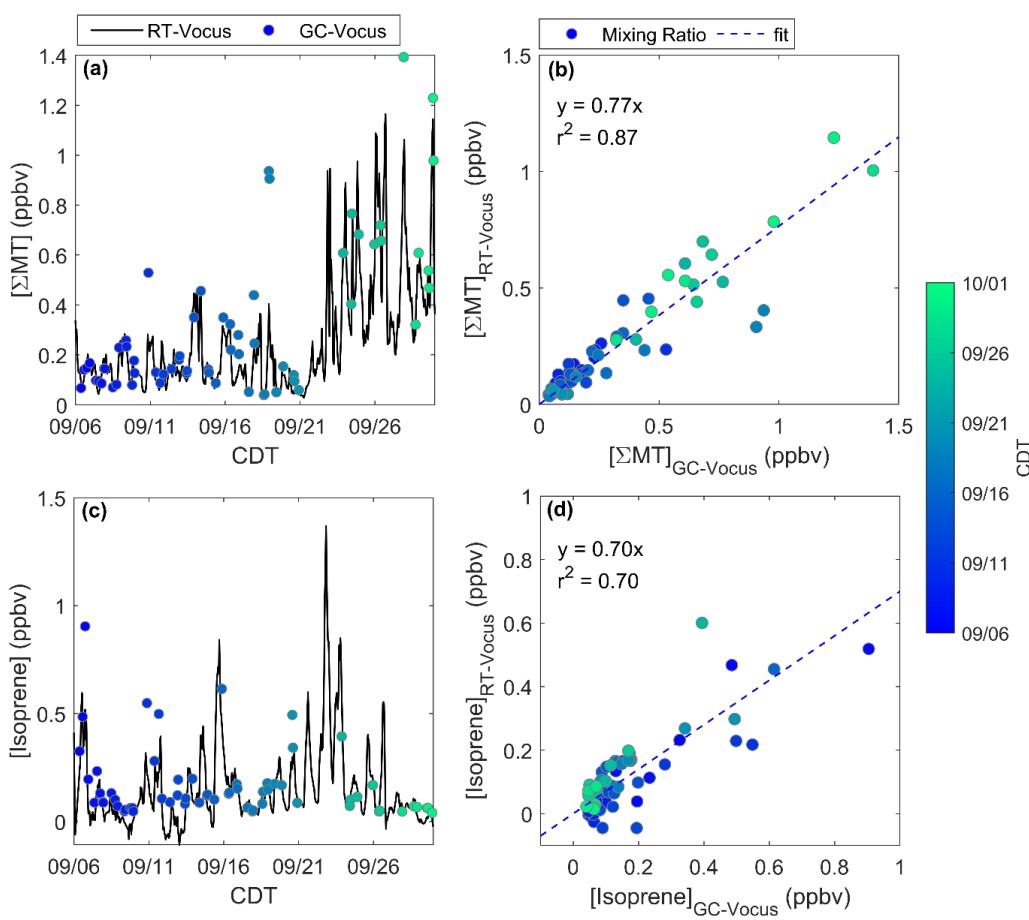


**Figure 6: Comparison of real time Vocus (RT-Vocus) and GC-Vocus for monoterpenes (a.) with a regression of calibrated RT-Vocus and GC-Vocus concentrations (b.). This is repeated for isoprene (c+d).**

GC-ToF observations of monoterpene oxides are presented in **Fig. 7**. The GC of $C_{10}H_{16}O$ (**Fig. 7a**) shows that the signal was primarily composed of three isomers very close in retention time (RetT = 620.3, 643.7, 672.1), corresponding to $RI_{obs}$ of 1091, 1120, and 1156. The first peak most closely matches in RI to $\alpha$-pinene oxide

($RI_{lit}$=1095) (Adams, 2000a), an epoxide product of $\alpha$-pinene ozonolysis (Alvarado et al., 1998) and was verified in lab post-study (**Fig. S12**). The second peak matches in RI to $\alpha$-campholenal ($RI_{lit}$= 1125) (Adams and Nguyen, 2005), a compound that has been observed as a product of $\alpha$-pinene oxidation in smog chamber studies (Jaoui and Kamens, 2003). The third peak matches in RI with trans-verbenol ($RI_{lit}$ = 1147) (Lucero et al., 2006), an anti-aggregation pheromone released by multiple bark beetle species (Lindgren and Miller, 2002) that are found in red pines in northern

WI (Pfammatter et al., 2015). Trans-verbenol has also been shown to be an $\alpha$-pinene oxidation product within the cells of Norway Spruce (Vaněk et al., 2005). It is also likely that camphor ($RI_{lit}$ = 1143) (Adams et al., 2006) makes





up the shoulder peak preceding the trans-verbenol peak, although potential error from the RI curve or differences in the published system vs. the GC-Vocus of this study may lead to camphor existing as the major peak. The RI of camphor also matched post-study (**Fig. S12**). Therefore, we label both peaks as "camphor/trans-verbenol". A 2D GC

study over the coniferous SMEAR II station in March-April 2003 recorded major $C_{10}H_{16}O$ compounds as camphor, $\alpha$-campholenal, and 4-caranone ($RI_{4\text{-caranone,lit}} = 1197$) (Kallio et al., 2006), which overlaps with two major $C_{10}H_{16}O$ compounds observed in the PEcoRINO study. Kallio et al. (2006) attribute observed $\alpha$-campholenal to oxidation of $\alpha$-pinene, 4-caranone to oxidation of 3-carene, and camphor to camphene oxidation. With a $\tau_{camphene+O_3}$ of 18 days and a $\tau_{camphene+OH}$ of 1.3 h under 51 ppbv $O_3$ (campaign max) and 4.0 x $10^6$ molecules $cm^{-3}$ OH (modeled campaign

max), respectively, the production of camphor would need to be the result of OH-initiated oxidation of camphene, if not directly emitted.

Chromatograms for two other monoterpene oxides, $C_9H_{15}O^+$ (**Fig. 7b**) and $C_{10}H_{15}O^+$ (**Fig. 7c**), had quantifiable peaks. The major peak in $C_9H_{15}O^+$ (RetT = 675.9 s; RI = 1159) matched best with nopinone, a predominant product of β-

pinene ozonolysis with an $RI_{lit} = 1138$ (Lucero et al., 2006). This peak was also confirmed to be nopinone in post-study GC calibrations (**Fig. S9**). A second peak was identified as 2,4 nonadien-1-al ($RI_{obs.} = 1187$, $RI_{lit} = 1184$) (Takeoka et al., 1996), although its sourcing is unknown. The first identifiable peak in the $C_{10}H_{15}O^+$ chromatogram is perillene ($RI_{obs.} = 1096$, $RI_{lit} = 1099$) (Adams, 2000b), a monoterpenoid that is produced by *Perilla frutescens*, a mint-like species native to Southeast Asia and India deeming it an unlikely source (Nishizawa et al., 1990), and is the

product of bioconversion of $\beta$-myrcene in oyster mushrooms (Krings et al., 2008). The second identifiable peak in the $C_{10}H_{15}O^+$ chromatogram is *p*-cymene-8-ol ($RI_{fit} = 1170$, $RI_{lit} = 1183$) (Adams et al., 2006), a product of the bioconversion of $\alpha$-pinene by bacteria in soils (Amiri, 2012), and a compound found in the oils of watercress, a species invasive to Wisconsin that can be found in lakes, streams, reservoirs, damp soil, and wetlands (WDNR, 2010). A third peak is tentatively identified in the $C_{10}H_{15}O^+$ chromatogram as thymol (RT = 760 s, $RI_{obs.} = 1264$) , a phenol derivative

of *p*-cymene that is found in various plants, the latter of which has been observed to be a product of $\alpha$-pinene oxidation by OH, $O_3$, and $NO_3$ (Gratien et al., 2011). This peak matched closest to thymol in post-study GC calibrations (RI = 1255) but the literature RI is 1290 (Adams, 2000b). As with the SQT GC, we cannot conclude if there was a change in speciation in any of the monoterpene oxides following leaf senescence since the major peaks eluted after 10 minutes and limited 20-minute chromatograms were collected. The other monoterpene oxides quantified with the RT-Vocus

($C_{10}H_{17}O_2^+$ and $C_{10}H_{17}O_3^+$) were not detected by the GC likely due to a RT for these species outside of the 20-minute collection time. To resolve these species in the future, we recommend a longer chromatographic separation time, a modified temperature program, or the use of a different chromatographic column (e.g. a polar column) optimized for these more oxidized species.

Background chromatograms for the molecules presented in **Fig. 5** and **7** are presented in **Fig. S15 and S18**.



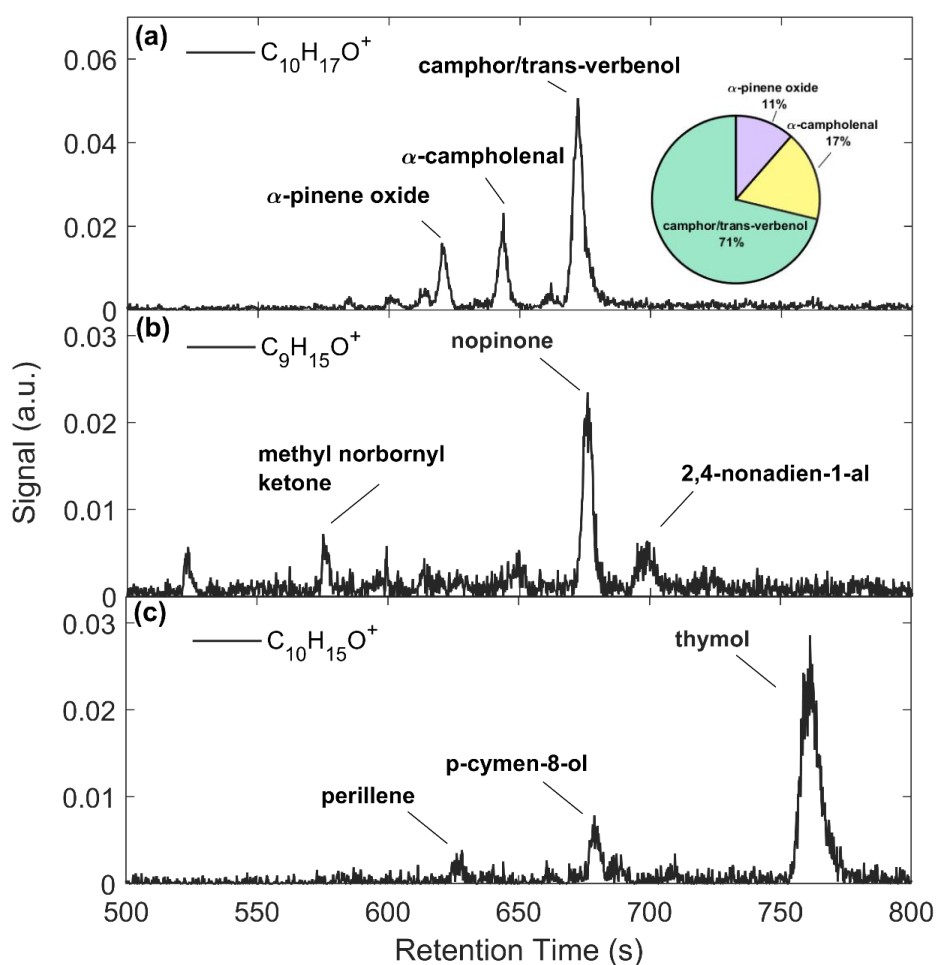

**Figure 7: a. Representative chromatograms of a. $C_{10}H_{17}O^+$, b. $C_9H_{15}O^+$, and c. $C_{10}H_{15}O^+$.**

## 4. Discussion

### 4.1 Controls of Observed Terpene Emissions

To determine the physical factors that control BVOC exchange at this site, flux observations were first compared to surface temperature to determine a light-independent temperature dependence. **Figure 8a** shows a regression of $F_{\Sigma MT}$ against temperature for the entire study (light grey circles). A portion of the data during the growing season (noted as periods before 21 September) are highlighted in dark grey circles and can be well fit by **Eq (3)**. This shows that the ecosystem emissions profile of MT prior to senescence followed a typical, consistent profile exponentially dependent

on leaf temperature. Here, we approximate leaf temperature as air temperature for both observations and parameterizations although the temperature of the canopy is likely different than ambient, with a recent study showing





an average ratio of $T_{canopy}$:$T_{air}$ of 1.03 and 1.07 for deciduous and evergreen species, respectively, at the temperate Harvard Forest (Still et al., 2022). The fit (red dashed line) results in a light-independent emission factor of 1.47 ppbv cm s$^{-1}$ (299 $\mu$g m$^{-2}$ h$^{-1}$) if we use an average loss factor within the canopy for all VOC ($\rho$) as 0.95, the latter value used

for this site in Vermeuel, et al. (2021). The fitted $\beta$ parameter (0.13) is close to the MEGAN MT $\beta$ (0.1) as well as experimentally determined $\beta$ (0.13) of a ponderosa pine tree in late August 2009 (Helmig et al., 2013).

This temperature dependence was also observed for $F_{C_{10}H_{16}O}$, as shown in **Fig. 8b**. The fit of **Eq (3)** provides a light-independent emission factor of 43.1 pptv cm s$^{-1}$ (9.8 $\mu$g m$^{-2}$ h$^{-1}$). The $\beta$ parameter derived from this fit (0.09) is close

to the value used for "other monoterpene" class in MEGAN (0.1) but deviates from the canopy-scale $C_{10}H_{16}O$-specific value measured over a coniferous forest in New England (0.21) (McKinney et al., 2011). In that same study the $\beta$ MT was 0.1 (close to this study's value of 0.13), suggesting that either camphor and MT in the New England study were the result of different biochemical processes or were from separate trees. From the Helmig et al. (2013) study, the $\beta$ of various terpenoids did not change significantly across six different pine species, so it is expected that the $\beta$ of both

MT and camphor would match other field studies if they were both derived from pine emissions, however this is not the case for $C_{10}H_{16}O$. This suggests that in our observations either $C_{10}H_{16}O$ and MT emissions come directly from the same species and follow biochemical pathways that have the same temperature dependence or $C_{10}H_{16}O$ is a secondary ambient product of MT emissions (i.e., in-canopy oxidation or oxidation below the sensor). We can consider the possibility of $F_{C_{10}H_{16}O}$ from in-canopy oxidation by estimating the required chemical lifetime of MT against oxidation

at this site. This is done using the equation for reactive chemical flux due to oxidation:

$$F_{product} = F_{precursor} \cdot k_{oxidation} \cdot C_{oxidant} \cdot \tau_{canopy} \cdot Y, \tag{4a}$$

where $F_{product}$ is the flux of an oxidation product due to in-canopy chemistry, $F_{precursor}$ is the flux of the reactant or precursor being emitted (i.e., MT), $k_{oxidation}$ is the bimolecular chemical rate constant for the oxidation reaction, $C_{oxidant}$ is the oxidant concentration, $\tau_{canopy}$ is the residence time of a parcel of air within the forest canopy, and $Y$ is

the product yield of the reaction. Using campaign-averaged fluxes of $C_{10}H_{16}O$ and MT (**Table 1**), an estimated $\tau_{canopy}$ of 5 minutes, and $Y_{\alpha-pinene\ oxide}$=0.02 from Alvarado et al. (1998), we can calculate the average lifetime of MT against oxidation required to reproduce observations:

$$\tau_{MT+oxidant} = (k_{oxidation} \cdot C_{oxidant})^{-1} = \frac{F_{precursor} \cdot \tau_{canopy} \cdot Y}{F_{product}} \tag{4b}.$$

The solution for **Eq (4b)** is 2 min, a value much lower than the lifetime of $\alpha$- or $\beta$-pinene against 28 ppbv $O_3$ (4.6 hr

and 1.1 day, respectively) as well as lifetime of $\alpha$- or $\beta$-pinene against $2.0 \times 10^6$ molecules cm$^{-3}$ OH (2.6- and 1.8-hr, respectively). The calculated $\tau_{MT+oxidant}$ in this study is on the lower end of the estimated $\tau_{BVOC+O_3}$ required to reproduce observed reactive fluxes of HCOOH from in-canopy ozonolysis at the same site in the summer of 2019 ($\tau_{BVOC+O_3}$~ 1-30 minutes) (Vermeuel, et al., 2020). Based on the longer chemical lifetimes of the identified MT species at this site, this finding suggests that only a small fraction of the observed $C_{10}H_{16}O$ emissions should be from in-

canopy MT oxidation.



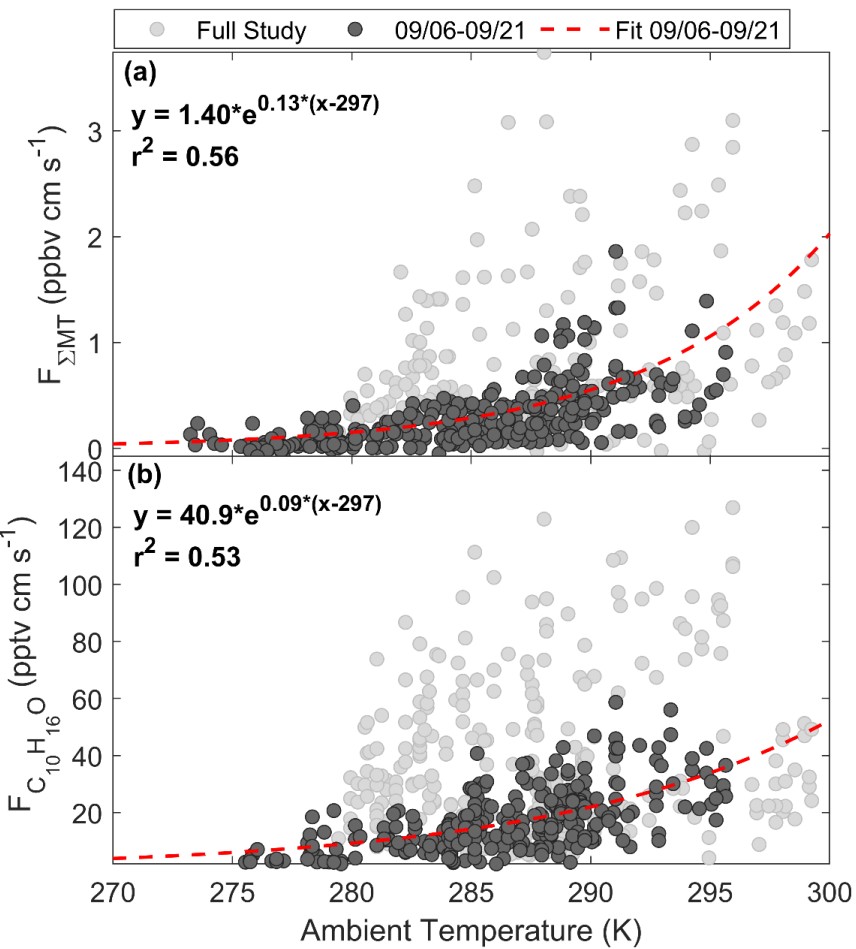

Figure 8: a. Regressions of $F_{\Sigma MT}$ and b. $F_{C_{10}H_{16}O}$ against ambient temperature for the entire PEcoRINO study (light grey circles) shows different exponential dependences of observed fluxes on temperature across the study.


**Figure 9a-d** presents time series of parameterized (red line) and observed (black line) fluxes of reactive terpenes. **Figure 9e-h** shows regressions of observations against parameterizations for the two leaf stage periods which are summarized in **Table 2**. For parameterizing fluxes of MT and $C_{10}H_{16}O$, we used the light-independent fits of flux vs. temperature from **Fig. 8** to get the temperature response factor β, and then incorporated the light-dependent fraction (LDF) of emissions to complete **Eq (3)**. For MT we used an LDF of 0.4 was used since it is the mean and median value among α- pinene (0.6), β-pinene (0.2), and the other monoterpene class (0.4) in MEGAN 2.1. For $C_{10}H_{16}O$ an LDF of 0.4 was used and for SQT we used a LDF of 0.5 and a β of 0.17 as per Table 4 of Guenther et al. (2012). The pre-exponential factors from **Fig. 8** were used for MT and $C_{10}H_{16}O$ and EFs for isoprene and SQT were based on best fits of the data.



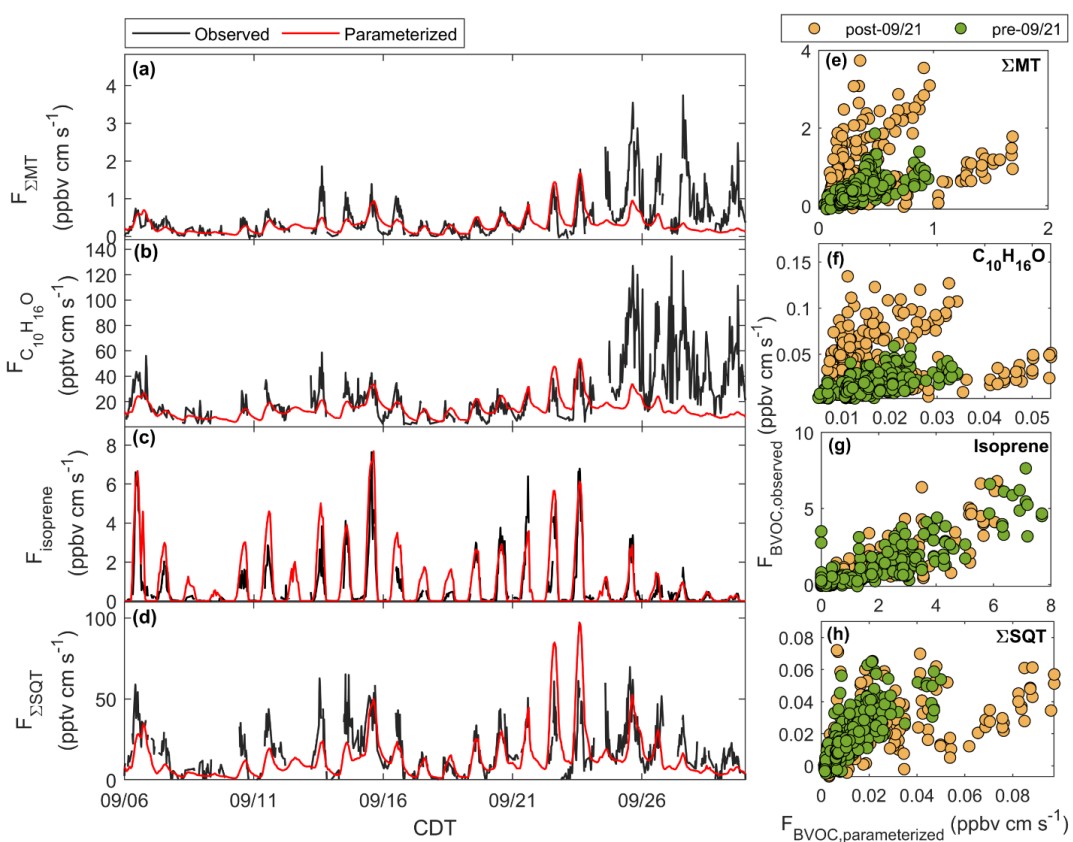

**Figure 9: Time series of observed $F_{\Sigma MT}$ (black line) along with parameterized $F_{\Sigma MT}$ (redline) from the fitted equation derived from 06-21 September is shown in a. and is repeated for b. $F_{C_{10}H_{16}O}$. c. The time series of observed $F_{isoprene}$ (black line) and parameterized $F_{isoprene}$ (red line) following E2. D. The time series of observed $F_{\Sigma SQT}$ (black line) and parameterized $F_{\Sigma SQT}$ (red line) following E3. Panels e.-h.: regressions of observed vs. parameterized fluxes for periods after (yellow circles) and before (green circle) 21 September.**

| Molecule | Obs. Vs. Param. Slope pre-09/21 post-09/21 | $r^2$ pre-09/21 post-09/21 | $\overline{F_{obs.}:F_{param.}}$ pre-09/21 post-09/21 |
|---|---|---|---|
| **ΣMT** | 1.2 | 0.52 | 1.2 |
|  | 0.46 | 0.058 | 2.1 |
| **C$_{10}$H$_{16}$O** | 1.4 | 0.44 | 1.2 |
|  | 0.13 | 0.0021 | 2.3 |
| **Isoprene** | 0.60 | 0.76 | 0.66 |
|  | 0.75 | 0.79 | 0.88 |
| **ΣSQT** | 1.4 | 0.62 | 1.5 |
|  | 0.32 | 0.24 | 1.0 |

**Table 2: Summary of slopes and correlation coefficients (r²) of observed vs. parameterized fluxes from Fig 9e-h. All fits are through the origin. Included are ratio of the averages of observed and parameterized fluxes during the pre- and post-09/21 periods.**



**Figure 9a** presents the time series of the parameterized $F_{\Sigma MT}$ (red line) following the fit of the 06-21 September data (**Fig. 8a**) overlaid on observed $F_{\Sigma MT}$ (black line). The MT EF used (299 $\mu$g m$^{-2}$ h$^{-1}$) is lower than the standard condition MEGAN 2.1 emission factors (EF) for the sum of for $\alpha$- and $\beta$-pinene (the majority of $F_{\Sigma MT}$) from a needleleaf evergreen temperate tree (800 $\mu$g m$^{-2}$ h$^{-1}$) and a broadleaf deciduous temperate tree (530 $\mu$g m$^{-2}$ h$^{-1}$) (Guenther et al., 2012). There is good agreement between the parameterized and observed data up until 24 September, after which the

emissions of MT can no longer be predicted by that temperature response curve. Since LAI changes significantly between 06-24 September (**Fig. 1d**) it would not be expected that this time series would be consistent, and it would be expected that in the later portions of the month the parameterized and observed $F_{\Sigma MT}$ would decrease. Since this is not the case, it is assumed that direct emissions of MT are not dependent on the observed change in LAI. The same behavior is observed for the time series of the parameterized $F_{C_{10}H_{16}O}$ overlaid on observed $F_{C_{10}H_{16}O}$ (**Fig. 9b**),

suggesting that $C_{10}H_{16}O$ arrives from the same observed LAI-independent source as MT. The EF for $C_{10}H_{16}O$ (9.8 $\mu$g m$^{-2}$ h$^{-1}$) is 5.4% of the value used for the "other monoterpenes" group in Guenther et al. (2012) for a needleleaf evergreen temperate tree (180 $\mu$g m$^{-2}$ h$^{-1}$), of which camphor would be considered part of.

    **Figure 9c** presents a time series of $F_{isoprene}$ along with the parameterized emissions following **Eq (2)**. LAI was

interpolated throughout the study period to scale emissions based on available leaf area. The data was best fit using a base emissions factor of 400 $\mu$g m$^{-2}$ leaf h$^{-1}$, which makes a range of emissions factors from 320-1600 $\mu$g m$^{-2}$ h$^{-1}$ for the month. This range is closer to the MEGAN standard conditions emission factor for needleleaf evergreen temperate trees (600 $\mu$g m$^{-2}$ h$^{-1}$) but lower than that of a broadleaf deciduous temperate tree (10000 $\mu$g m$^{-2}$ h$^{-1}$). Excellent agreement between modeled and measured isoprene fluxes is achieved by using a low base emissions factor and a

dynamic LAI (**Fig. 8g, Table 2**), suggesting that the measured flux of the ion $C_5H_9^+$ is predominantly due to isoprene. It also further supports that the satellite-derived LAI is more representative of deciduous, isoprene-emitting trees rather than MT-emitting coniferous species.

    Emissions of SQT can be estimated by **Eq (3)**, however, the net flux of SQT may be underestimated due to in-canopy

oxidation prior to escaping the canopy. Reduction in net emissions can be corrected by applying **Eq (4)** to the parameterized emissions. The within canopy loss term, ρ, from **Eq (3)** is ignored in this calculation. **Figure 9d** shows the result of the parameterized flux of $\beta$-farnesene (red line), using an optimized EF of 30 $\mu$g m$^{-2}$ h$^{-1}$, a value 10 $\mu$g m$^{-2}$ h$^{-1}$ less than the EF of $\alpha$-farnesene used in Guenther et al. (2012), that is corrected for in-canopy loss due to ozonolysis and OH-initiated oxidation. When calculating reactive loss, it is assumed the [O$_3$] measured at 30 m was

the same within the canopy and that [OH] followed the same profile as the modeled OH (**Sect. 2.5**). Accounting for in-canopy chemistry reduced the flux by 10%, on average, for the whole study period. This parameterized and corrected $F_{\beta-farnesene}$, is in good agreement with observations prior to 21 September of the study (r$^2$ = 0.62), and poor agreement following 21 September. This shows that although there is no observed enhancement of SQT emissions in the latter portion of the study, the leaf senescence period does show a deviation from what is expected

based on parameterizations.





### 4.2 Potential mechanisms of BVOC enhancement during leaf senescence

The increased and sustained $F_{\Sigma MT}$ and [ΣMT] throughout late September as well as the change in MT speciation suggests that the mechanism of MT emissions changes throughout the summer to autumn transition at this site. While the exact mechanism is still unknown, we provide a few potential reasons for this enhancement along with their likelihood of contribution. Briefly they are: 1) a reduction in ambient oxidant loading that slows chemical loss, 2) increased contribution of MT from leaf litter or soils, and 3) physical changes to plants and modifications to terpene synthesis during senescence.

#### 4.2.1 Changes in ambient oxidation chemistry

Reduction in gas-phase oxidant concentrations would reduce the rate of in-canopy oxidation and could increase the net detected flux. To estimate how a change in-canopy oxidation would impact measured $F_{MT}$, we can calculate a net, detected flux ($F_{MT,net}$) by taking the difference of the emitted flux ($F_{MT,emitted}$) and the sum of oxidation reactions:

$$F_{MT,net} = F_{MT,emitted} - \Sigma(F_{MT,emitted} \cdot k_{oxidation} \cdot C_{oxidant} \cdot \tau_{canopy}) \quad \textbf{(5a)}.$$

Rearranging gives the fraction of $F_{MT,net}$ to $F_{MT,emitted}$ which can be used to assess in-canopy loss under different oxidant loadings:

$$\frac{F_{MT,net}}{F_{MT,emitted}} = 1 - \Sigma(\tau_{oxidation}^{-1} \cdot \tau_{canopy}) \quad \textbf{(5b)}.$$

Since $O_3$ increases after 21 September, the concentration of OH would need to decrease after 21 September for this to increase the net emitted MT, as observed. Using observed $O_3$ concentrations and a high estimate of $1 \times 10^7$ and $5 \times 10^6$ molecules cm⁻³ OH before and after 21 September, respectively, and a $\tau_{canopy}$ of 5 minutes, the ratio of $\frac{F_{MT,net}}{F_{MT,emitted}}$ would increase from 0.79 to 0.89 before and after 21 September, a modest increase. Since the parameterized $F_{MT}$ (which is an estimate of $F_{MT,emitted}$) (**Fig. 9a**) is a factor of 1.5 higher in the period before 21 September relative to after, this change due to loss by oxidation would be small compared to change in direct emissions from temperature and sunlight. Still, a reduction in photochemically produced OH from attenuated solar radiation in the latter parts of the month (**Fig. 1e**) likely has a small impact on sustained [ΣMT].

#### 4.2.2 Contributions from soil and leaf litter

An explanation for the high observed $F_{\Sigma MT}$ and [ΣMT] may be enhanced emissions from the forest floor during leaf senescence and abscission. Hakola, et al. (2003) observed high concentrations of MT on the same order or higher than those observed in peak growing season (July) at a primarily pine boreal forest in Hyytiälä, Finland over two years. This may be due to the fact that throughout autumn the forest floor can be a significant contributor to VOC flux from the decomposition of leaf litter (Isidorov et al., 2010; Greenberg et al., 2012) or from microbial activity in exposed soils (Mäki et al., 2019). Aaltonen et al. (2011) observed forest floor BVOC emissions in the same forest in Hyytiälä peaking in early summer and autumn with emissions of MT averaging 5.04 $\mu$g m⁻² h⁻¹ and containing primarily $\alpha$-pinene, camphene, and $\Delta^3$-carene with a negligible contribution from isoprene and SQT. Hellén et al. (2006) recorded forest floor emissions from the same site, noting that the highest MT emissions occurred in the spring and autumn with values ranging from 0-373 $\mu$g m⁻² h⁻¹ and composed of $\alpha$-pinene, camphene, $\Delta^3$-carene, limonene, and $\beta$-pinene.




However, this behavior has been observed primarily in boreal forests when needleleaf litter is high and may not be reflective of the CNNF at the stage in needleleaf cycle observed in the PecORINO study where most of the needle leaves were still on the trees. This would also conflict with observations of SQT at the site which showed no day-to-day variability and should also have been controlled by conifers. Bare soils can serve as a source of monoterpenes in temperate regions, although that contribution ($\sim$82 ng m$^{-2}$ h$^{-1}$) is generally negligible (Trowbridge et al., 2020)

compared to foliar and leaf litter emissions. However, soils can contain a significant amount of camphene (Hayward et al., 2001) which may contribute to the early autumn change in speciation observed in this study. Based on these studies we assume that decaying leaf litter and soils may provide a small contribution to the early autumn rise in $F_{\Sigma MT}$ as well as the change in MT speciation during the summer to autumn transition. This study shows a need for measurements of forest floor emissions in temperate regions or in mixed forests that can confirm the magnitude and

speciation of these emissions.

### 4.2.3 Leaf degradation or enhanced reactive carbon synthesis

A third, and most likely, explanation for elevated MT emissions is the leaf senescence process itself. Leaf senescence is a catabolic process that redistributes nutrients to seeds or newly developing organs to ensure optimal production of offspring (Lim et al., 2007). During stages of senescence, changes in the biomechanical properties of the epidermis

cuticle make diffusion of certain hydrophobic compounds easier through a degraded epidermal layer, which could lead to increased emissions of BVOC (Mozaffar et al., 2018). Degradation of plant structural components can also generate leaf wounds during senescence, which may enhance MT emissions if the lamina is wounded and significantly increase MT flux if the midrib is cut (Portillo-Estrada and Niinemets, 2018). In addition, the degradation of cells provokes desiccation of the lamina, driving the emission of volatile compounds stored within the cytosol or in

specialized reservoir organs (Portillo-Estrada et al., 2020). These two processes have been witnessed in trees of the genus *Populus*, including aspen, which make up 25% of the acreage of CNNF as of 1996 (Haugen et al., 1998).

A study measuring VOC emissions via the eddy covariance method over a poplar plantation of 12 different genotypes in the second half of 2015 observed a peak in the emissions of 25 VOCs at the beginning of September and the onset

of senescence (Portillo-Estrada et al., 2020), with increases in OVOCs (methanol, acetone, acetaldehyde, MT alcohol, MVK+MACR) of 1-2 orders of magnitude and modest increases in MT. The authors also observed a cessation of isoprene flux beginning with the onset of leaf senescence (and loss in canopy greenness) due to reduced photosynthetic activity. This agrees with observations at CNNF with increases $F_{\Sigma MT}$ and $F_{C_{10}H_{16}O}$ coincident with sharp drops in $F_{isoprene}$ near to 21 September. This also supports using LAI, or leaf greenness, to scale parameterized $F_{isoprene}$ and

generate agreement in parameterizations and observations. In addition, this explanation is consistent with the observation that $F_{SQT}$ in this region are controlled by pines that did not yet undergo senescence and abscission during this measurement period. Various *Populus* species at a CNNF site have been observed to emit small amounts of MT, although there are many other potential tree species at CNNF that can emit more MT than poplars (e.g., northern white cedar, *Thuja occidentalis*; balsam fir, *Abies balsamea*; white spruce, *Picea glauca*). Still, if aspens within the flux

footprint were controlling BVOC emissions, it would be expected, according to the Portillo-Estrada et al. (2020) study,




that other OVOC such as methanol or acetone would present an enhancement in emissions, which was not the case. The observed fluxes of methanol and acetone throughout the study with post-09/21 fluxes exhibited a net sink (**Fig. S19**). It is possible that the net flux of those OVOC were controlled by other enhanced routes of deposition such as uptake to water films or biotic processes (Laffineur et al., 2012), or that there is enough genotypic variation between the cited study and this one that makes it difficult to assess changes in the gross source of methanol and acetone.


A final enhancement route related to the senescence process may be due to increased synthesis of, and need to mitigate, reactive oxygen species within leaves. During senescence, reactive oxygen species (ROS) are generated in response to stress which depletes antioxidants and damages cells (Jajic et al., 2015). To reduce ROS, plants may increase

synthesis of reactive carbon as has been observed in some flowering plants where internal concentrations of isoprene were increased to protect against singlet oxygen ($^1O_2$), an ROS generated during senescence (Affek and Yakir, 2002). It is known that MT serves as ROS scavengers during senescence (Chen and Cao, 2008) and production of MT in response to increased oxidative stress has been shown in the leaves of evergreen oak (*Quercus ilex*), similar to what has been observed for isoprene antioxidative production (Loreto et al., 2004). Although there are no studies confirming

enhanced MT synthesis in senescing plants at this site, there is the possibility for this biochemical pathway to be upregulated during this phenological stage and future studies aimed at assessing plant reactive carbon synthesis in response to senescence and oxidative stress would be invaluable.

**4.3 Photochemical box model calculations of HOM and H₂SO₄ production**

The presence of MT, MTOs, and DMS raises the question of whether MT or DMS contributes more to aerosol production following BVOC oxidation at this forested site and what the impact of enhanced MT emissions following leaf senescence has on aerosol production. To answer these questions, we use the 0D box model described previously that evaluates terpene chemistry to produce HOM and DMS chemistry to produce H₂SO₄.

Model solutions of $P_{HOM}$ and $P_{H2SO4}$ are presented in **Figure 10**. **Fig. 10a** presents $P_{HOM}$ calculated from separate model conditions that use observed $F_{\Sigma MT}$ (black line) as well as the parameterized $F_{\Sigma MT}$ presented in **Fig. 9a** (red line). Also included is the ratio of calculated $P_{HOM}$ using observed $F_{\Sigma MT}$ and $P_{HOM}$ using parameterized $F_{\Sigma MT}$ presented as a blue dashed line. This ratio shows that $P_{HOM}$ is underestimated by as much as a factor of 6 in a day when using parameterized MT flux and for the post-09/21 period and $P_{HOM}$ is on average 2.2x larger in the observationally

constrained model. The $P_{HOM}$ value can be used as a proxy for the mass transfer of gas to a transition regime particle and thus the production of organic aerosol mass (and aerosol growth) if it is assumed that the mass accommodation of HOM to particles is always unity and the average particle radius is similar throughout the study (Fuchs and Sutugin, 1971). Therefore, there is a significant underestimation in estimated organic aerosol mass during the onset/continuation of the senescing canopy if a model uses a common parameterization for MT emissions and having

more comprehensive plant trait data that correlates with different stages of senescence in combination with empirical knowledge of BVOC emissions during senescent phases is key for improving estimates of organic aerosol production.



**Figure 10b** shows $P_{H_2SO_4}$ from a model run that considers DMS as the only source of $H_2SO_4$ (black line) as well as a model run with DMS and estimated $SO_2$ (gold line) from a regional $SO_2$ monitor. Since [DMS] throughout the study showed no seasonal dependence, $P_{H_2SO_4}$ does not change significantly throughout the study, with a study average of $2.3 \times 10^6$ molecules cm$^{-3}$ day$^{-1}$. The US-PFa site is approximately 12 km east of a paper mill in Park Falls, WI and 75 km northwest of a paper mill in Rhinelander, WI which provides a large source of $SO_2$ and is the site of an EPA $SO_2$ monitor. Since the observed [DMS] is very low (<10 pptv), it is possible that some days experience outsourced $SO_2$ that controls onsite $P_{H_2SO_4}$. The potential impact of this is addressed by running the model with [$SO_2$] = 250 pptv (as a lower end from the EPA monitor), giving a study $P_{H_2SO_4}$ average of $5.9 \times 10^8$ molecules cm$^{-3}$ day$^{-1}$. This implies that if plumes advected from a regional paper mill were to consistently intersect with the US-PFa site for a prolonged period, then the outsourced plumes would primarily control $H_2SO_4$ and inorganic particle production.

**Table S4** outlines model solutions of $P_{HOM}$ and $P_{H_2SO_4}$. For the case where there is no anthropogenic influence (i.e., no outsourced $SO_2$), $P_{HOM}$ as constrained by observations is more than a factor of 250 higher than $P_{H_2SO_4}$ when averaged over the study period. For the case where we assume constant outsourced $SO_2$ from a paper mill $P_{HOM}$ and $P_{H_2SO_4}$ are the same value ($5.9 \times 10^8$ molecules cm$^{-3}$ day$^{-1}$) when average over the study period. From these modeled estimates we can conclude that biogenic particle formation and growth in this region is largely controlled by HOM production, specifically by MT oxidation.

This analysis provides a first step in approximating the impact of BVOC on particle nucleation or growth from gas-phase HOM production, although detailed conclusions cannot be made without actual aerosol measurements that allow for a more complete calculation of nucleation and condensation rates. Calculated concentrations would be used in a parameterization of neutral biogenic particle nucleation rates (in the absence of ions) of 1.7-nm mobility diameters ($J_{n1.7}$, cm$^{-3}$ s$^{-1}$) for $H_2SO_4$ (Kirkby et al., 2011) and HOM (Kirkby et al., 2016). Additionally, the chemical mechanism used in this study does not evaluate the impact of $SO_2$ stabilization of Criegee Intermediates from alkene ozonolysis which would enhance $Y_{HOM}$ from $O_3$ for BVOC in the high $SO_2$ case (Stangl et al., 2019). Future measurements should include monitoring of particle size and number to determine the specific rate of mass transfer of gaseous HOM and $H_2SO_4$ to the particle phase as well as ambient $SO_2$ and OH to better calculate $H_2SO_4$ production and validate the employed mechanism of DMS oxidation.

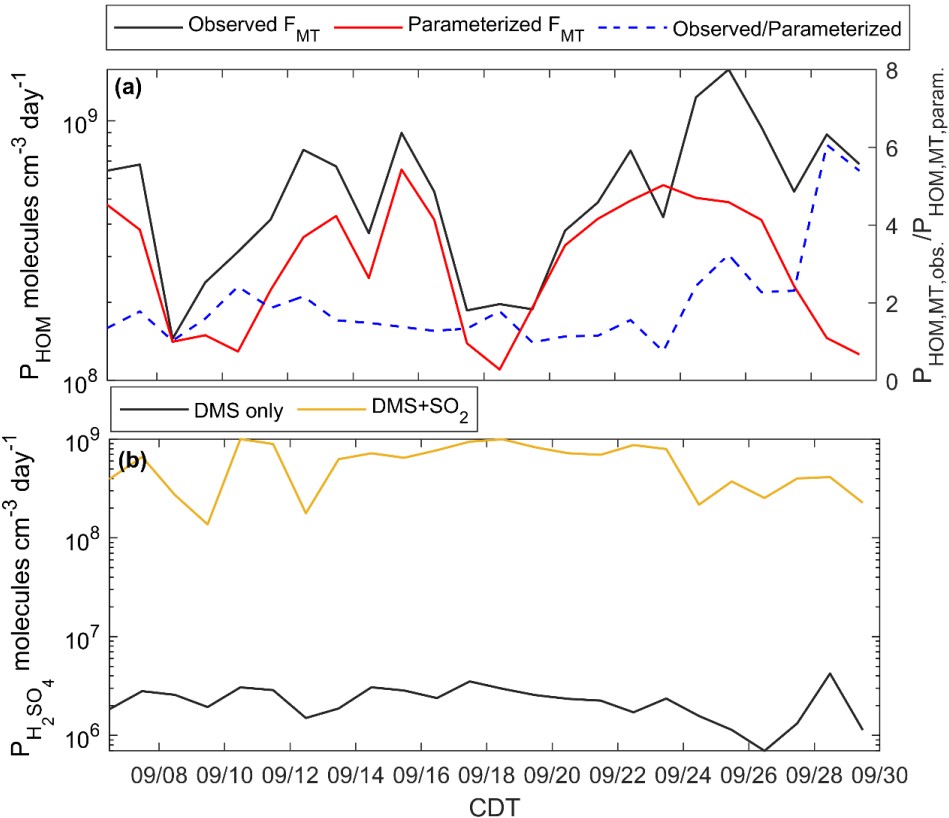

**Figure 10: Modeled HOM and H₂SO₄ during the PEcoRINO study. a. Calculated $P_{HOM}$ using observed $F_{MT}$ (black line) and parameterized $F_{MT}$ (red line) shows that $P_{HOM}$ is underestimated by as much as a factor of 6.1 when calculating the ratio of the two production rates (blue line). (b.) $P_{H_2SO_4}$ as determined from DMS only (black line) and DMS and regional $SO_2$ (gold line) show highest production between 17-24 September. The addition of 250 pptv $SO_2$ increases $P_{H_2SO_4}$ by over an order of magnitude, making $P_{H_2SO_4}$ comparable to $P_{HOM}$.**

## 5 Conclusions

Northern temperate forests experience a steep change in physical canopy conditions throughout the summer to autumn transition that can make predictions of canopy-mediated BVOC exchange difficult. This study measured dominant BVOCs throughout the month of September to assess the response of the forest canopy in mediating reactive carbon to a wide range of meteorological conditions and multiple stages of the leaf life cycle. Results from this study showed that the observed concentrations and emissions of MT and a monoterpene oxide ($C_{10}H_{16}O$) may be enhanced: 1) primarily due to the senescence process that degrades leaves as well as increases ROS and potentially antioxidative reactive carbon, and with additional contribution from 2) the emissions of the forest floor or freshly decomposing leaf litter, and/or 3) a decrease in the oxidants that remove MT (e.g. $O_3$, OH, $NO_3$) within or above the canopy, allowing for enhanced emissions and sustained concentrations due to longer MT lifetimes. The flux of $C_{10}H_{16}O$ is primarily from direct emissions although a small fraction could be from in-canopy oxidation of MT. Multiple monoterpene



oxides were observed on site and GC confirmed their identities to be a combination of MT oxidation products and directly emitted compounds. The observed flux of MT can be well parameterized following an exponential

temperature dependence up to the estimated start of leaf abscission (around 21 September) indicating that global models can well replicate the diurnal cycle and relative magnitudes of MT emissions during the growing season but may fail to do so during leaf senescence and onward. In this region emissions of isoprene exceed those of MT and can be well-replicated following a common parameterization based on light and temperature and scaled by LAI, suggesting that most of the leaves lost were isoprene-emitting while living. Speciation of MT *via* GC showed that MT was

primarily composed of $\alpha$- and $\beta$-pinene and the change in season shifted the distribution of MT to majority $\beta$-pinene. This shift in speciation may be driven by some MT being removed from the collective profile due to senescence, allowing other coniferous compounds that are sustained to now become relatively more dominant. To our knowledge, we present the first recorded canopy-scale fluxes of SQT in a mixed temperate forest, with the main isomer identified as $\beta$-farnesene. DMS was also observed onsite in low quantities (~10 pptv) with no dependence on time of month.


A box model incorporating measured terpenoid flux and DMS and $O_3$ concentrations and inferred OH and $NO_3$ concentrations show that the production of HOM is underestimated by as much as a factor of 6.1 when incorporating common parameterization of MT flux. Further, biogenic particle formation and growth should be dominated by organic (terpene-derived HOM) rather than inorganic (DMS-derived sulfate) constituents although the incorporation

of anthropogenic $SO_2$ brings HOM and $H_2SO_4$ production to be the same average value. Results from this study highlight the need to consider leaf senescence in mixed forests where emissions of reactive BVOC can be enhanced and calls for more observations of BVOC exchange in northern temperate forests during the summer to autumn transition. This would allow us to determine if the observed role of the canopy before and after the onset of leaf abscission follows a seasonal cycle or if the observations from PEcoRINO were anomalous to this region and time.

This study also shows that DMS sources in a northern temperate mixed forest composed of woody wetlands are low and should be a small contributor to regional aerosol production, with the majority of aerosol formation and growth predicted to be from terpene oxidation or oxidation of outsourced anthropogenic sulfur.

**Data availability**

Concentrations and fluxes of ambient species presented in this study can be found at

http://digital.library.wisc.edu/1793/83610. US-PFa meteorological data can be found at https://doi.org/10.17190/AMF/1246090.

**Supplement**

Additional GC methods, cospectra, flux quality control details, and supporting figures and tables.



**Author contributions**

MPV and THB designed the research. THB and ARD supervised the project. MPV carried out ambient Vocus sampling and analyzed the data. GAN, JT, MSC, and BML assisted with ambient deployment. JT collected meteorological data and PAC provided $O_3$ measurements. DBK conducted laboratory calibrations and assisted in GC characterization. MSC, BML, and AMT aided in interpretation of data. MPV wrote the manuscript. All co-authors reviewed and edited this manuscript.

**Competing interests**

At least one of the (co-)authors is a member of the editorial board of Atmospheric Chemistry and Physics. The peer-review process was guided by an independent editor, and the authors also have no other competing interests to declare.

**Acknowledgements**

The authors would like to thank Jeff Ayres of the Wisconsin Educational Communications Board for his assistance at WLEF. G. M. Wolfe is gratefully acknowledged for publicly providing FluxToolbox and F0AM (archived on GitHub), a MATLAB base of analysis scripts, portions of which were altered for use in this analysis. Hariprasad Alwe and Dylan Millet are acknowledged for providing the sampling inlet used in this study. We acknowledge that this project occurred on the traditional territory of the Ojibwe people.

**Financial support**

This work was supported by National Science Foundation (NSF) Grant GEO AGS 1822420 and AGS 1829667. Flux observations at US-PFa were supported by the US Department of Energy Ameriflux Network Management Project award to the ChEAS core site cluster and the NOAA Carbon Cycle and Greenhouse Gases tall tower program. Patricia Cleary acknowledges support from the University of Wisconsin – Eau Claire's Blugold differential tuition fund for faculty-student collaborative research and NSF Grant GEO AGS 1918850.

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
