# Peer review of "Observations of biogenic volatile organic compounds over a mixed temperate forest during the summer to autumn transition"

_EGUsphere, 2022_

## Author Comment (AC1)

*We thank the reviewers for their constructive feedback and detailed review of our manuscript and have made revisions based on their suggestions. Below are the suggestions with our responses in blue italics.*

**Reviewer 1**

The manuscript by Vermeuel et al. presents an interesting BVOC dataset wherein the emission and mixing ratios of monoterpenes and its oxidation products are enhanced from the onset of senescence/abscission. They also have coupled a Vocus-PTR-MS with GC to speciate and confirm the identity of different isomers/compounds presented in the manuscript. I think this is one of the first EC measurements using the Vocus PTR-MS and would recommend publishing this manuscript after some issues that I noted are addressed.

"Specific comments": The manuscript is comprehensive but lacks details on the instrumental and measurement setup of the experiment along with some other issues.

1.  Length of main sampling line is not mentioned. The operation pressure and field strength inside the ion source of the instruments would be of interest for users who would use the instrument for similar EC studies. Include these in Section 2.1.3.

    *Thank you for pointing this out, these details should have been mentioned. The sampling line length has been added on line 144. We have also added other helpful instrument specs in lines 150-153 as per your suggestion:*

    *"The focusing ion–molecule reactor (FIMR) of the Vocus was held at 1.5 mbar and the FIMR front and back were held at 400 and 35 V, respectively. The Vocus big segmented quadrupole (BSQ) ion guide was maintained at 215 V to allow higher transmission of lower molecular weight molecules such as methanol."*

2.  There is mention of calibration every 4 hours but no mention of instrument zeros. Where there zero measurements done routinely (more frequently) for the setup and subtracted from measurements?

    *Thank you for this comment. Zeros were performed with each calibration and it was important that we mentioned this.*

    *Line 158 now says:*

    *"Zeros were also performed during calibrations."*

3.  If zeros were done, didn't the zero subtraction help with isoprene correction?

    *Thanks for this comment and the following one regarding clarification on correcting $C_5H_9^+$ for isoprene. It shows that we needed to do a better job explaining how isoprene was calculated. We have provided a calculation in **S3** and explain more here and in the response to the next comment.*

    *The n-aldehyde signal is from ambient so it would show no signal during a zero and is not used for a correction. In other words, the n-aldehyde signal also zeroed.*

*We determine the contribution of n-aldehydes to the $C_5H_9^+$ signal ($C_{5H9,n-ald}^+$) using the ratio of $C_{5H9,n-ald}^+$ to the parent n-aldehyde signal ($M_{n-ald}^+$) as determined by the GC peak areas of the aldehyde isomers. We justify using this fragment:parent ratio for the RT-Vocus since it is robust for alpha pinene ($C_6H_9^+$:$C_{10}H_{17}^+$, Figure S6) and should translate for fragmented species. We also can assume that these n-aldehydes (C7,C8,C9), along with isoprene, are the primary species in the $C_5H_9^+$ signal based on the GC (Fig. 5). We then retrieve isoprene in the RT-Vocus as the difference between the total measured $C_5H_9^+$ signal and that derived from n-aldehyde fragmentation.*

$$C_5H_9^+{}_{n-ald,RT} = M^+{}_{C7ald,RT} \cdot \frac{C_5H_9^+{}_{C7ald,GC}}{M^+{}_{C7ald,GC}} + M^+{}_{C8ald,RT} \cdot \frac{C_5H_9^+{}_{C8ald,GC}}{M^+{}_{C8ald,GC}} + M^+{}_{C9ald,RT} \cdot \frac{C_5H_9^+{}_{C9ald,GC}}{M^+{}_{C9ald,GC}}$$

$$C_5H_9^+{}_{isoprene,RT} = C_5H_9^+{}_{total,RT} - C_5H_9^+{}_{n-ald,RT}$$

In Fig 2, isoprene mixing ratios after correction (from aldehyde contribution) is negative. How did the RT-Vocus aldehydes signal behave? Where the signal always present during measurements like in GC-Vocus or did it have any pattern? The clarification for the above comment would be helpful here. The correction of isoprene (line 349-350) could be better written as an equation.

*The sum n-aldehyde signal is included in Fig. S7 but also below for ease of review. The signal was relatively persistent with concentrations lowest in the morning (6-8 CDT). The approach we take for isoprene correction is justified since we only seek to correct out n-aldehyde signal from C5H9+ and not determine its ambient concentrations.*

[Figure]

*We agree that a more detailed description of this correction is necessary, and we have added a new section on this calculation and its uncertainty in the supplement, S3. We have clarified this paragraph to show that we are looking at GC peak area ratios and have added lines 361-363 to say:*

*"**S3** describes this correction and associated uncertainties in more detail with uncertainties calculated from accuracies in calibrant standards and mass flow controllers as well as $1\sigma$ uncertainty of isoprene calibration factors and the GC peak area $C_5H_9^+$ to $M^+$ ratio".*

4. What % of dataset passed the EC quality criteria? Was integral turbulence characteristics test not done?

   *Thank you for this comment. It showed us that we needed a more rigorous treatment for assessing turbulence. We have updated with more rigorous tests for turbulence using the test for integral turbulence characteristics and included these criteria in S2. 51% of data did not pass EC quality criteria. We have also updated lines 239-240 to say:*

   *"Post-field quality control removed 51% of measured flux periods."*

5. MEGAN v2.1 emission factors ($\varepsilon$) are standardized also for a LAI of 5 along with T and PAR (pg 3183 Guenther et al. 2006). Eq 2 and 3 must include LAI: more importantly here since the satellite LAI decreased rapidly from 4 to 0.8. This is also needed if you want to compare the $\varepsilon$ that you obtain with MEGAN $\varepsilon$ later in section 4.1. Since the MT emission enhancements go opposite to LAI, your results would be more imperative. I suggest redoing the parametrizations including LAI. There are python, MATLAB and excel MEGAN versions for a single measurement site for more straightforward comparisons.

   *Thanks for pointing this out. We did scale by LAI for isoprene but you are correct that scaling of LAI should be used for all species if we are comparing to routine predictions of emissions. We have incorporated a gamma LAI value for all fluxes which is used in MEGAN to correct standard emissions to actual LAI. Gamma LAI is a correction factor that equal 1 when LAI = 5. This is how MEGAN corrects for LAI in CTMs such as GEOS-Chem.*

   *We have updated equations 2 and 3a to include this and have included line 249 to say:*

   *"… $\gamma_{LAI}$ is a correction factor for LAI where $\gamma_{LAI}$=1 at an LAI of 5…"*

   *Since $\gamma_{LAI}$ is now included in our parameterizations fitted to observations pre-09/21, we have removed discussions about LAI dependences pre-09/21 throughout the text.*

   How is ρ, the loss factor within canopy calculated?

   *Thanks for the question. We point to a citation that describes this (Vermeuel, et al., 2021) but it is much later in the text and there should have been a citation earlier on. ρ is calculated follow equation 21 in Guenther et al 2006*

   $$\rho = 1 - \frac{D}{\lambda \cdot u_* \cdot \tau + D}$$

   *where D is average the canopy depth (15 m), λ is a fitted parameter (0.3), and τ is the average chemical lifetime of the compound (2 h), resulting in a ρ of 0.95. We added in the Guenther 2006 citation in line 244 to have a full citation of the equations we use.*

6. Line 425 says footprints overlap but source area is shifted. But a flux footprint plot is missing. Since the forest is not homogeneous it would be good to show if the footprint has not changed drastically before and after 21st September. Are there more high MT emitters in southwest?

*We agree that a footprint analysis is required and has been added to the SI (Fig. S9c+d) to support this. The footprint analysis is also shown below. Footprints before and after 09/21 show a shift in the prevailing footprint direction but the average 70% footprint covers similar regions. We are saying that if there were higher emitters in the southwest we would have observed that in the earlier parts of the month at times when winds came from that direction. However, since there was no notable increase in flux in that direction during the 09/06-09/21 period (Fig. S9a+b), this does not support the hypothesis that there was an increase due to a change in source.*

[Figure]

*Lines 439-442 have been revised to say:*

*"While there is a source area shift for $F_{\Sigma MT}$ from the west half to primarily southwest for pre- and post-21 September, respectively, it is unclear if this shift caused emissions enhancements since both footprints overlap according to flux footprint prediction (FFP) parameterizations (Kljun et al., 2015) (**Fig. S9**)."*

7. The GC results obtained here are very interesting but can take off attention from the main results. I suggest moving Section 3.2 to SI.

*Thank you for this suggestion. We agree that too much focus on the GC section takes away from the main ideas concerning changes in flux over the seasonal transition. However, this combined flux/GC study using this Vocus system is the first of its kind and the GC results aid in the interpretation of contributions to observed BVOC flux. This combined data set presents a novel and detailed description of reactive BVOC over this specific ecosystem and would be helpful for future studies at this routinely used flux site. Further, an explanation and correction of the isomer-resolved contributions to C5H9+ is critical for not only this study but for PTRMS users that previously considered C5H9+ to be primarily isoprene.*

*To address the concern, we have shortened all parts within Section 3.2 and moved some technical details to the SI to maintain focus on the key parts of the GC contributions.*

8. Did you observe any vertical flux for DMS? Jardine et al., 2015 has seen similar mixing ratios like Brown et al., 2015 but have also reported emissions from trees and concluded that the emissions could be from both soil and plants. Recent studies have also shown trees to be a source of DMS (Vettikkat et al. 2020). How did you conclude the DMS observed cannot be from trees?

*Thanks for pointing this out. Since the signal was so low no flux was observed for DMS. We concluded that DMS could not be from trees because of its very low observed abundance (<10 ppt) and because none of the plants in the cited works were within our footprint. However, without leaf level or soil chamber measurements we cannot conclude either source definitively. Further, the Berresheim and Vulcan, 1992 citation observed similar mixing ratios to this study and concluded most of the DMS mass was coming from the crown of lolblolly pine. We have corrected lines 378-380 to say:*

*"There was no dependence of [DMS] on leaf stage or LAI, suggesting that DMS may not be sourced from plants or are from plants that did not show a change in LAI."*

*Lines 383-391 also now say:*

*"Temperate coniferous ecosystems can have DMS sourced from trees. Vertical distributions of DMS in a loblolly pine forest near Atlanta, Georgia also showed enhanced [DMS] closer to the forest floor and at night (~12 pptv) compared to the day (~4 pptv) (Berresheim and Vulcan, 1992), with abundances similar in magnitude to this study. The authors of the Georgia study attribute this distinction to reduced photooxidation at night and concluded that DMS emissions were from the pine trees. However, soil emissions which, although highly dependent on microorganisms in the soil, have been proven to provide a small source in other ecosystems (Goldan et al., 1987; Banwart and Bremner, 1975; Yang, 1996) and can also explain the magnitude of observed mixing ratios at the site. Without leaf-level or soil chamber measurements of DMS we cannot definitively state whether DMS comes from the soils or trees"*

9. Flux plots are crowded with error bars. Change the plot with shaded error bands for better readability. Also edit color palette of plots according to color blindness guidelines.

*Plots have been changed with shaded error bands and color-blind palettes along with different shapes/line styles for monochromatic color blindness.*

10. Abstract could be edited to highlight the main findings as given in the conclusions.

*Thanks for your suggestion. We have edited the abstract to better set up the purpose of the study and show the conclusions that:*

1. *We saw an enhancement in MT and MT oxide fluxes as well as a shift in MT speciation*
2. *Parameterizations cannot simulate fluxes of MT, MT oxides, and SQT post-21 September but can reproduce isoprene*

3. *Senescence likely has a role in terpene emissions*
4. *HOM production is underpredicted using parameterizations compared to observations.*

"Technical comments."

1. Line 230: fx(t) is only defined in SI.

   *This has been removed replaced by "the cross-covariance"*

2. Line 326: 'higher' instead of 'high'

   *Corrected.*

3. Line 480: Check if it is RIfit or RIobs

   *This was changed to RIobs. Other instances of RIfit were also replaced by RIobs.*

4. Line 491: Fig 6c should be Fig 6b
5. Line 521: Fig 6b should be Fig 6c
6. Line 673: make the D in caption small: Use parathesis (d)) to reduce confusion.
7. Line 700: Fig 8g should be Fig 9g

   *These changes have all been made.*

8. Table S2 & S3: alphas and betas missing
9. Table S3: Most headers missing
10. Table S4 caption and title has text missing

   *These three are formatting errors when converting to PDF. Our apologies for overlooking this. These have been corrected.*

**Reviewer 2**

The manuscript describes an interesting study that measured BVOC concentrations and fluxes above a mixed coniferous/deciduous forest. One unique aspect was using a GC system to speciate monoterpenes and other BVOCs with the PT-TOF-MS. Another interesting point was considering BVOC emission during leaf senescence. The methodology is sound and well described in the manuscript.

I have a number of specific comments that need to be addressed. Most are minor, but there are several major items that need to be addressed. First, the measurements show an interesting enhancement of ecosystem-level emissions of monoterpenes after leaf senescence. But the authors go too far with the data they have collected and present too much speculation as results. I provide a number of detailed comments on this concern below. Next, the inclusion of the DMS data, while again interesting, makes the paper too long and does not have a strong

enough scientific connection to the rest of the material. The paper will be improved by removing this information and perhaps it can be included in another publication.

Lines 138-139: give the inner diameter of the tubing.

*This has been added. Lines 136-137 now says:*

*"…an inlet composed of Type 1300 Synflex (3/8" ID)…"*

Line 159: "these values" is a bit confusing. Instead, state explicitly "the calibration factors".

*Thanks for pointing this out and we agree. "these values" was changed to "the calibration factors".*

Lines 175-176: there is no trouble with water vapor at this temperature?

*Thanks for the comment. The system was optimized to have a low enough trapping temperature where water issues were avoided. This temperature was 20 °C and water condensation is avoided due to compensating effects such as the pressure drop across the oxidant trap. We have edited lines 177-179 to say:*

*"Both the sample collection and focusing are conducted at sub-ambient (20 °C; optimized to avoid condensation) temperatures through the use of a Peltier thermoelectric cooler."*

Lines 181-182: was there a backflush to remove heavier HCs from the column during the 10-min runs?

*Yes, the system backflushes the columns at an elevated temperature, 25 °C over the maximum column temperature, after the elution of targets is complete.*

Line 235 & SI lines 44-54: when computing the lag time for each 30-min period, was there still a clear max at night?

*Thanks for your question, this is a good point. The max at night was not clear which is why we included a quality control filter for lag time. If lag times were outside of the prescribed range, they were considered erroneous. However, this still might not be strict enough since you can potentially have a noisy max within the QC lag window. Ideally this would also be removed from flux LoD calculations since your max will be in the noise at that point. Either way, as per suggestions from Reviewer 1, we have applied more rigorous quality control filters for assessing turbulence which has removed most of the nighttime data.*

Line 248/Eq 3a: Shouldn't gamma sub P only be applied to the LDF term? Also, give a bit more detail. This equation seems to be the leaf-level Guenther 1995, not the canopy emission model MEGAN, Guenther 2012. Are you accounting for leaf area, etc?

*Thanks for your questions regarding this. The equation here is derived from equations 1 and 2 in Guenther 2012 and assumes emission factors at standard conditions where LAI = 5. This was not clear the way it was written before because there was no consideration of LAI. We have now included and LAI term using $\gamma_{LAI}$ where $\gamma_{LAI}$=1 at LAI = 5, which is written out in Guenther 2006 and cited.*

*The activity factor, γ, treats deviations from standard conditions and here is the product of $\gamma_{LAI}$, $\gamma_P$, and $\gamma_T$ where $\gamma_P = (1 − LDF_i) + LDF_i\gamma_{P\,LDF}$ and $\gamma_T$ is calculated as described in the text. This is how MEGAN 2.1 is written in the cited Guenther et al 2012 paper. We only show $\gamma_T$ as separate terms because we use the temperature dependent fit later in the text but point to Guenther 2012 for full equations of other γ. We have revised the text in this section as per the suggestions of Reviewer 1 and incorporated $\gamma_{LAI}$.*

Lines 318-320: Also, could be changes in the intensity of vertical mixing, which in turn could be influenced by changes in the structure of the canopy due to leaf loss. Should add this and discuss.

*Thanks, this is a good point. We do not believe that a decrease in vertical turbulent mixing is responsible for enhanced mixing ratios since there was no change in intensity of turbulent vertical mixing, as calculated by $\sigma_w/U$, pre- and post- 09/21 as shown in the time series and median diel cycle below.*

[Figure]

[Figure]

*Since we discuss the causes for MT enhancement in detail in 4.2 we have decided to remove this section and not provide additional discussion here.*

*Lines 327-328 now say:*

*"We observe an increase in $\Sigma MT$ concentrations following 21 September, most likely due to senescing leaves, as we will discuss in **4.2**."*

Lines 352-353: This is a really big correction. You need to do an assessment of the error it introduces into your isoprene measurements.

*We agree that this correction is a significant portion of the $C_5H_9^+$ signal and describe it here to show other PTRMS users that this correction may be needed in other instruments. We have included the night and day ppt correction here to further clarify this correction and show the absolute changes to the isoprene concentration. A 59% correction is large but this is because isoprene is low at night and aldehydes are persistent.*

*To clarify this correction lines 360-361 now say:*

*"The contribution of n-aldehydes made up 36% (148 ppt correction) and 59% (140 ppt correction) of the daytime and nighttime $C_5H_9^+$ signal, respectively."*

*The uncertainty in the correction factor is relatively small given our ability to constrain the fragmentation of n-aldehydes using the GC. We have included more detail in SI section S3 and addressed this for Reviewer 1. Briefly, to calculate isoprene concentrations from $C_5H_9^+$ we need to calculate the $C_5H_9^+$ signal from n-aldehydes which is the product of the n-aldehyde parent ion $M^+_{n-ald,RT}$ and the GC-derived ratios of $\frac{C_5H_9^+_{n-ald,GC}}{M^+_{n-ald,GC}}$. We then correct out the aldehyde contribution and calibrate for isoprene using the isoprene calibration factor. The uncertainty in isoprene raw signal is mainly driven by errors in zeroing and potential loss to the inlet which we determined as ~1%, the uncertainty in n-aldehyde signal in $C_5H_9^+$ depends on variability in $\frac{C_5H_9^+_{n-ald,GC}}{M^+_{n-ald,GC}}$, and the uncertainty in the isoprene calibration factor relies on accuracy in standards and mass flow controllers as well as the variability in the field-determined calibration factor. So the isoprene mixing ratio is calculated as*

$$[isoprene_{RT}] = \frac{C_5H_9^+_{total,RT} - C_5H_9^+_{n-ald,RT} (\pm unc_{n-ald,RT})}{calfactor (\pm unc_{calfactor})}$$

Since

$$C_5H_9^+_{n-ald,RT} = M^+_{C7ald,RT} \cdot \frac{C_5H_9^+_{C7ald,GC}}{M^+_{C7ald,GC}} + M^+_{C8ald,RT} \cdot \frac{C_5H_9^+_{C8ald,GC}}{M^+_{C8ald,GC}} + M^+_{C9ald,RT} \cdot \frac{C_5H_9^+_{C9ald,GC}}{M^+_{C9ald,GC}}$$

Its uncertainty is:

$$unc_{C_5H_9^+_{n-ald,RT}} =$$
$$\sqrt{(\sigma_{C7ald,ratio} \cdot M^+_{C7ald,GC})^2 + (\sigma_{C8ald,ratio} \cdot M^+_{C8ald,GC})^2 + (\sigma_{C9ald,ratio} \cdot M^+_{C9ald,GC})^2}$$

$$\approx$$
$$\sqrt{(\frac{\sigma_{C7ald,ratio}}{C7ald,ratio} \cdot C_5H_9^+_{C7ald,GC})^2 + (\frac{\sigma_{C9ald,ratio}}{C9ald,ratio} \cdot C_5H_9^+_{C9ald,GC})^2 + (\frac{\sigma_{C9ald,ratio}}{C9ald,ratio} \cdot C_5H_9^+_{C9ald,GC})^2}$$

*which has an average value of 14.4 cps and a maximum of 49.0 cps corresponding to ≈ 17.0 and 54.0 pptv, respectively.*

*The uncertainty from the isoprene calibration factor was calculated to be +/- 11.8% which results in an average uncertainty in $[isoprene_{RT}]$ of 17.1% during the day and 35.8% at night. This uncertainty is applied presented in section 3.1.2 and shown in the figures as shaded areas.*

I haven't seen this correction in other PTM-MS papers. Do you have any references?

*Thanks for the comment. To date there are no other references showing this, further urging the need to share this information with other PTRMS users. By using evidence from the GC and RT Vocus in the field and lab we have provided sound reasoning for using this correction. Although not yet published, we know of at least one group that has used this correction in urban and commercial regions where the concentrations of n-aldehyde > isoprene.*

Can you quantify how much was from the Na2SO3 trap?

*The correction here is only for signal from the RT-Vocus. We only use the peak area ratios of the GC to convert from M+ to C5H9+ since $\frac{C_5H_9^+{}_{n-ald,GC}}{M^+{}_{n-ald,GC}} \approx \frac{C_5H_9^+{}_{n-ald,RT}}{M^+{}_{n-ald,RT}}$ as shown in the α-pinene case (Fig. S6).*

*We cannot quantify how much n-aldehyde was from the $Na_2SO_3$ trap, but we do not need to know how much n-aldehyde was detected by the GC since the peak area ratios of $\frac{C_5H_9^+{}_{n-ald,GC}}{M^+{}_{n-ald,GC}}$ is due to fragmentation and will always stay the same.*

Line 420: remove "physical." These also could be biological and/or chemical factors. Also, switch all units, especially for Fig. 4, to the mass-based flux units which are the convention in the field. All the literature you site is in mass/(area x time) units: be consistent with the existing literature.

*Thanks for the suggestion. The word "physical" has been removed. We agree that the units of "ppb cm/s" are not straightforward units of flux since it is a value of mass/(area x time). However, it is also conventional to use molar and molecular units instead of mass which we find are clearer for emissions inputs and subsequent chemistry calculations in atmospheric chemical models and for interpretation of our box model results. Because of this, we have changed all units to molecules $cm^{-2}$ $s^{-1}$ and converted any references to have the same units for ease of comparison.*

Lines 423-425: you haven't discussed any footprint analysis. Since this statement is inconclusive, simply remove it.

*We have added in a footprint analysis as per the comments from Reviewer 1. This statement is now supported and we will keep it in the text.*

Line 441: insert "presumed" before cessation, since you did not actually measure this, but are inferring it from leaf senescence.
*"Presumed" has been added.*

Line 443-446: Note that having a flux observed at mass 69 that behaves like isoprene does not

mean you have successfully corrected the concentration at mass 69 with your correction. Even if there is still a significant offset in 69 due to another interference, if that offset is not correlated with the vertical wind speed, it will not contributed to the flux. Look back at your original Eq. 1. In theory, you could not apply the correction, and you would get the same flux, again as long as the interference is not correlated with w. Note this isn't exactly true, since some of the corrections you apply might be influenced by the absolute concentration of mass 69.

*The way this was explained in the text is unclear. The isoprene flux was calculated without an n-aldehyde correction because of potential uncertainties introduced to the calibrated concentration through this correction. We support the omission of this correction on lines 446-447 in the submitted manuscript where we state that there was no measurable flux from any of the main aldehydes that interfered with $C_5H_9^+$ with the implication that if the aldehyde parent masses have no flux, then those species should not affect the $C_5H_9^+$ flux and are rather just an offset not correlated with w. However, to stay consistent with all methods we should include the corrected isoprene signal as the flux. Either way the results are nearly identical because the aldehydes do not vary with w. Below is a figure comparing flux derived from uncorrected and corrected $C_5H_9^+$ colored by $C_5H_9^+$ signal from n-aldehydes.*

[Figure]

*Additionally, we have rewritten lines 461-465 as*

*"…implying that parameterizations of isoprene emissions based on sunlight and temperature are appropriate during this season. Since there was no measurable flux from the parent masses of heptanal, octanal, and nonanal we are confident that there is minimal to no added error from corrections to the isoprene signal since the aldehyde signals do not vary with w and therefore should not contribute to $C_5H_9^+$ flux."*

Line 449: "an" not "and".
Line 491: This is panel 6b, not 6c, which is isoprene.

*Thank you for catching these. These have both been corrected.*

Lines 490-492: while I agree this is good agreement between the two methods, why is the real-time data lower? Since the GC method only considers three compounds, shouldn't it be the lower value?

*This is a valid point. Aside from the uncertainty in GC observations, part of this issue could be that we only had in-field calibrations for alpha-pinene for the RT-Vocus data while there were three observed isomers. These isomers could have different fragmentation patterns which lead to different calibration factors at the parent ion $C_{10}H_{17}^+$ as well as lower signal at $C_{10}H_{17}^+$ that is then calibrated as alpha-pinene.*

*While we cannot correct this with lab data now, we do need to address this potential cause for disagreement. Lines 510-512 now say:*

*"Since the RT-Vocus ΣMT was calibrated in the field using only α-pinene we can suspect that the underreporting of RT-Vocus concentrations from $C_{10}H_{17}^+$ is partially from higher fragmentation patterns of the other MT isomers."*

Also, an inefficiencies with the trap would lead to the GC data being lower.

*Since we calibrated the GC peak areas frequently for isoprene and monoterpenes, this would account for trap inefficiencies.*

Line 494-524: I understand that many issues arise in field work, and often it is necessary to perform post-field experiment corrections and laboratory tests to recover data that was compromised by unexpected processes. But given the magnitude of the correction (1/3 during the day) for isoprene and the large variation in Figure 6d, the error bars on the isoprene concentration should be over 50%.

*We have now addressed this in an earlier comment. The average error for isoprene concentrations is 17.1% during the day and 35.8% at night.*

(Note that you refer to 6b on line 521, but like above on line 491, that appears to be swapped and should be 6c.) While r2 is over 0.7, I don't consider that very good to start with and also much of that fit is driven by one high point. At lower isoprene concentrations (< 0.4 ppbv), there is only a poor correlation, visually.

*Thanks for the comment. Your concern for representativeness of fit is a valid one so we have added to the best fits a bootstrapped 95% confidence interval analysis that picks 500 sets of random subsamples and fits those. This is now included in Figure 6 and 8 and pointed out in the text in sections. Lines 509-510 now say:*

*"Bootstrapped confidence intervals of the slope provide a range of 0.66 to 0.86."*

*And lines 528-531 have been updated to say:*

*"There is good agreement between the data ($r^2$ = 0.67, slope = 0.68), with bootstrapped confidence intervals of the slope providing a range of 0.57 to 0.84. This shows that our $C_5H_9^+$ correction method is a viable solution for calculating only isoprene from $C_5H_9^+$ and would be useful for other studies where fragments may contribute to a portion of a signal."*

*For isoprene the range in $r^2$ for these solutions primarily fall in the 0.6-0.7 range with a mean of 0.66 which is still a good agreement.*

[Figure]

*Additionally, for MT the majority $r^2$ values are above 0.84.*

[Figure]

Fortunately, as discussed above, you can have isoprene concentration errors that do not influence your flux calculation. But, you need to conduct a more rigorous error analysis for the isoprene concentrations, and give error ranges whenever the concentrations are presented. This includes visually in graphs.

*We agree that a more rigorous error analysis was needed and thank you for suggesting it. As discussed above, error has been propagated now and the correction to isoprene is now detailed. Time series figures have uncertainty shading and uncertainty numbers are listed. Lines 319-323 state:*

*"Uncertainties for all mixing ratios are presented as shaded regions and were calculated by propagating uncertainty from fraction lost to the inlet (Fig. S1), accuracy in calibration standards and mass flow controllers, and the standard deviation (1σ) in field calibrations. This produced*

*average uncertainties of 11.1, 29.0, 12.0, and 8.0 % for ΣMT, ΣSQT, DMS, and O₃, respectively. The average isoprene uncertainty was 17.1% during the day and 35.8% at night and is further discussed in this section."*

*Lines 413-416 state:*

*"Since there was no in-field calibration for monoterpene oxides, we approximate uncertainties for $C_9H_{15}O^+$, $C_{10}H_{15}O^+$, and $C_{10}H_{17}O^+$ to have the same relative uncertainty as ΣMT and $C_{10}H_{17}O_2^+$, $C_{10}H_{17}O_3^+$ to have the same relative uncertainty as ΣSQT based on assumptions of volatility".*

Lines 617-620: need more detail about the regression. You refer to Equation 3, but you have equations 3a and 3b in the text. Maybe your regression is only for temperature, while Equation 3a has light? I think maybe that's the case, but you need to be more explicit here. It's also confusing, since you have an exponential fit in Fig. 8a but also mention the loss factor rho. But note that the loss factor, if it's a simple exponential fit, won't affect the beta term.

*Our apologies for the confusion in presentation. In this section we are looking at what environmental factors control terpene emissions to produce a parameterization similar to MEGAN. We first fit flux data to a simple exponential fit against temperature to show that there is a predicted temperature dependence up until 09/21 and then this dependence falls off after the onset of senescence, an interesting finding by itself which we explain.*

*Lines 626-652 have been reworded and condensed to get to the point concerning MT and MT oxide temperature dependence. We mentioned rho for readers to make a comparison to MEGAN emission factors (since emission factor = pre-exponential fit/rho) but have removed the mention of rho here since it is not the purpose of these paragraphs and will be used in the full parameterization later.*

*We use the fit to get a beta factor and pre-exponential factor to use in the full parameterization of equation 3a to compare to observations (Fig. 9). The beta factor remains the same in our full parameterization, but the pre-exponential factor describes just the light-independent emission factor. We want the full emission factor used in 3a to better factor in the light, temp, and LAI dependencies so we derive an emission factor that best fit the pre-09/21 data using the pre-exponential temperature dependence factor as an initial guess. The full parameterization is then compared to observations. In our initial submission we did this for just MT and C10H16O but have now performed a fit for isoprene and SQT for consistency. The main takeaway is that the enhancement in MT and MT oxide BVOC emissions cannot be described by commonly used parameterizations. We do not use the exact emission factors from MEGAN because the purpose is not to show that the absolute values of MEGAN are wrong but rather cannot reproduce seasonal transition phenomena. We compare the growing season-derived emission factors to MEGAN emission factors in similar ecosystems to show how they deviate for this specific footprint as a point of interest but not the main conclusion.*

Line 622: assuming air temperature is necessary but problematic. The reference Still et al 2022 is very good, but in parenthesis include "deg C" [using actual degree sign!] since the units matter with this ratio.

*Thanks for pointing this out. We have added (in °C).*

Lines 638-655: there is a _lot_ of existing literature about in-canopy oxidation which you are ignoring. Here is one citation: https://acp.copernicus.org/articles/12/8829/2012/. But you need to incorporate a state-of-the-art understanding into your discussion.

*We have decided to remove this section because it pulls focus from this subsection that is discussing environmental factors and comparisons to parameterizations. We bring up this same concept and set of equations in the manuscript in 4.2.3 so we modified the discussion there based on your suggestion about in-canopy oxidation literature. Our full response can be found under the second comment on in-canopy oxidation.*

Lines 663-669: you need more detail on how this was accomplished. See comments above where I have experienced confusion about references to Eq. 3.

*We agree that this was not written clearly. We have addressed this fully in a previous comment. Lines 628-635 now explain this more and say:*

*"We parameterized fluxes of MT and $C_{10}H_{16}O$ using β and $\epsilon$ derived from the fits in* **Fig. 8***. The parameterizations use observed β as a constraint and $\epsilon$ as an initial guess to best fit pre 21 September data to* **Eq (3a)***. For MT we used an LDF of 0.4 was used since it is the mean and median value among α- pinene (0.6), β-pinene (0.2), and the other monoterpene class (0.4) in MEGAN 2.1. For $C_{10}H_{16}O$ an LDF of 0.4 was used and for SQT we used a LDF of 0.5 and a β of 0.17 as per Table 4 of Guenther et al. (2012). Parameterized emission factors for isoprene (***Eq 2***) and SQT (***Eq 3***) were based on best fits of the pre-21 September data. We use observed PPFD and T as well as satellite LAI as constraints for parameterizations. For terpenes other than SQT we apply an average loss factor within the canopy (ρ) of 0.95, a value used for this site in Vermeuel, et al. (2021)."*

Table 2: why have both the slope and the ratio of the fluxes? This information is largely redundant unless you have a specific reason to explore it.

*Thanks for the comment. We agree that including both the slopes and ratios are redundant. Since we have now fit all data to pre-09/21 conditions, there is not a need for comparing slopes between pre- and post- 09/21. The more useful values are the $r^2$ and the comparison of average values before and after 09/21 to determine how representative our pre-09/21 fit is as well as highlight the disparities in parameterization and observation post-09/21.*

Lines 701-702: while this might be true and is interesting, it is out of the scope of the current paper and the statement should be removed.

*This has been removed.*

Lines 716-802: This section is very speculative and is stretching the data you collected too far. You are making too many qualitative assumptions and pronouncements. This section is also very long and needs to be more focused on the data you have and the conclusions you can draw from it directly. You can speculate about one or two hypotheses in your discussion, but this section ranges too far from the data you have collected.

*Thanks for your comments on this section. We agree that some points are more speculative, and we have edited and cut down this section to focus more on the data we have and to point out what are our best guesses which we feel strengthens this section. The responses below address each specific comment on this section.*

Section 4.2.1: First, see my comment about the in-canopy oxidation literature. Second, you would need an error analysis to give ranges on your estimates.

*Thanks for the comment and we understand this concern. The purpose of this section is to provide parameterized scenarios where in-canopy oxidation would have the strongest impact. We believe that incorporating a 1D model treatment is beyond the scope of this analysis and not necessary for highlighting the main conclusion which is: after applying these conditions you still cannot explain the change in emissions due to a change in oxidant loading. We agree that we should explain this more and acknowledge that this is not a replacement for explicit calculations of concentration tendencies that consider transport, emissions, deposition, and chemistry.*

*Lines 692-696 now say:*

*"The approach used here is a simple parameterization and not a replacement for more comprehensive 1-D vertical models such as the Canopy Atmospheric Chemistry Emission model (Bryan et al., 2012) where changes in concentration with time are explicitly treated. However, since we use conservative estimates of $\tau_{oxidation}$ and a fixed $\tau_{canopy}$ we are directly comparing scenarios where in-canopy oxidation would have the strongest impact."*

Section 4.2.2: You don't have enough data to rule this out. Specifically:

Lines 752-753: I don't think canopy scale SQT measurements are of high enough quality to support this contention.

*We have removed this statement.*

Lines 753-755: you don't present any measurement data about the state of the soils. You are introducing qualitative, and perhaps observational, data into your discussion.

*We agree that there is no quantitative data on soils. We are only presenting previous studies and their magnitudes and speciation. This section has been condensed and we added a section that points this out and calls for more soil and floor studies.*

*Lines 711-713 now say*

*"However, without soil and floor observations we cannot conclude quantitatively their impact on observed $F_{\Sigma MT}$. This study shows a need for measurements of forest floor emissions in temperate regions or in mixed forests that can confirm the magnitude and speciation of these emissions."*

Lines 756-760: you need to be much more cautious in your conclusions.

*Agreed. See above comment for how we corrected this.*

Lines 764-766: From what I see, the Mozaffar reference discusses hydrophilic compounds. Please clarify.

*The cited paper does have some hydrophobic species, but in the cited study monoterpenes were below the detection limit and the rest were primarily hydrophilic. We have revised lines 715-718 to say:*

*"During stages of senescence, changes in the biomechanical properties of the epidermis cuticle make diffusion of certain compounds easier through a degraded epidermal layer, which could lead to increased emissions of BVOC, although this has been primarily recorded for hydrophilic species (Mozaffar et al., 2018)."*

Lines 788-790: again, very speculative. I would be very, very surprised if there was a genotypic variation. The emissions of methanol in particular are fundamental to plant biochemistry.

*We agree that we cannot comment on genotypic variation without speculation but the control of water films on highly soluble gases is a well understood process. This is particularly true for methanol and is contained within the Laffineur citation. We have future plans to use a suite of soluble OVOC from this dataset to evaluate a 1D model that treats the processes contained within Laffineur study and feel that explaining this part of the data set is beyond the scope of this paper. We bring in the methanol and acetone data since our cited paper supporting an enhancement in VOC emissions due to senescence also observed an enhancement in OVOC. However, we did not see this methanol/acetone enhancement potentially because our site was very humid and may not be directly comparable for water soluble species. Below is a figure of exchange velocities (Flux/concentration) of methanol and acetone showing a strong dependence of these OVOC on RH. Correcting for enhancements in conservative estimates of dry deposition due to increased stomatal uptake under high RH does not flatten this curve. We have included an additional reference that describes the control of OVOC on bidirectional flux.*

[Figure]

*Lines 732-735 have been revised to say:*

*"It is possible that the net flux of those OVOC were controlled by other enhanced routes of deposition such as uptake to water films or biotic processes (Laffineur et al., 2012; Fulgham et al., 2020) that makes it difficult to assess changes in the gross source of methanol and acetone."*

Lines 791-802: and again, very speculative. It is very difficult to understand leaf biochemical processes from whole-stand flux measurements.

*We agree that we cannot make definitive statements on biochemical processes and do not have leaf-level data to support that. We have shifted focused to more on desiccation and poplar emissions since we have data to compare against and say that this increase in ROS is speculative. We still believe this should be mentioned, with caution, to suggest future studies to better understand if this route actually influences emissions enhancements in seasonal transitions.*

*Lines 737-738 now say:*

*"A final, but most speculative, enhancement route related to the senescence process may be due to increased synthesis of, and need to mitigate, reactive oxygen species within leaves."*

Lines 823-850: see my summative comments, but while this is very interesting, it feels jammed into the current manuscript.

*Thanks for your comment on this section. We have shortened this section to only show the main points and rewrote the introduction paragraph of this section to transition to this section better. We would like to keep this section since the impact of enhanced emissions on the production of*

*aerosol precursors is a valid point of discussion. We also feel the manuscript abstract and introduction set up the paper to discuss this topic.*

Lines 861-866: your study did not demonstrate any of these mechanisms. Your study _did_ demonstrate that MT emissions were enhanced during senescence. You then speculate about mechanisms to explain this enhancement, but you cannot say your study demonstrated anything about these mechanisms.

*To avoid definitive statements of qualitative assumptions we have rewritten lines 794-798 to now say:*

*"Using qualitative assumptions of leaf and forest floor stage as well as estimations of the ambient oxidative environment, we propose possible causes to be: 1) the senescence process that degrades leaves and potentially increases antioxidative reactive carbon, 2) emissions of the forest floor or freshly decomposing leaf litter, and/or 3) a decrease in the oxidants that remove MT (e.g. $O_3$, OH, $NO_3$) within or above the canopy."*